# Theoretical insights into the Peierls plasticity in SrTiO$_3$ ceramics via dislocation remodelling

Yi Li[1,2,4], Xiangyang Liu[1,4], Peng Zhang[3], Yi Han [1], Muzhang Huang[1] & Chunlei Wan [1] ✉

An in-depth understanding of the dislocations motion process in non-metallic materials becomes increasingly important, stimulated by the recent emergence of ceramics and semiconductors with unexpected room temperature dislocation-mediated plasticity. In this work, local misfit energy is put forward to accurately derive the Peierls stress and model the dislocation process in SrTiO$_3$ ceramics instead of the generalized stacking fault (GSF) approach, which considers the in-plane freedom degrees of the atoms near the shear plane and describes the breaking and re-bonding processes of the complex chemical bonds. Particularly, we discover an abnormal shear-dependence of local misfit energy, which originates from the re-bonding process of the Ti-O bonds and the reversal of lattice dipoles. In addition, this approach predicts that oxygen vacancies in the SrTiO$_3$ can facilitate the nucleation and activation of dislocations with improvement of fracture toughness, owing to the reduction of average misfit energy and Peierls stress due to the disappearance of lattice dipole reversal. This work provides undiscovered insights into the dislocation process in non-metallic materials, which may bring implications to tune the plasticity and explore unknown ductile compositions.

In contrast to metals and alloys with excellent plasticity and fracture toughness, most ceramics and semiconductors are brittle at low-to-moderate temperatures, which has become a major concern in their applications. The main reason is that these non-metallic materials rarely have active dislocations at room temperature or lower temperatures. The Peierls stress, a measure of the slip resistance overcome by dislocations is always too large for these non-metallic materials due to the complex ionic and covalent bonds. However, in a few ceramics such as SrTiO$_3$[1–8] and MgO[9–12], the Peierls stress could be small enough to enable the easy motion of dislocations, leading to the enhancement of fracture toughness and even a certain degree of plasticity. Particularly, there has been a recent discovery of ductile ceramics and semiconductors at lower temperatures due to the easy sliding of

dislocations[13–15]. Therefore, an in-depth understanding of the dislocation dynamics in non-metallic materials and the corresponding Peierls stress has become increasingly important for improving their mechanical properties and identifying new ductile compositions.

For metallic materials, the motion of dislocations has been well described by the Peierls–Nabarro (PN) model[16–18], which offers a theoretical framework for modeling the dislocation core and the Peierls stress. In the PN model, the dislocation core is regarded as a continuous distribution of shear $S(x)$ or infinitesimal dislocations with density $\rho(x)$ [$\rho(x) = \mathrm{d}S(x)/\mathrm{d}x$], where $x$ is the coordinate in the glide plane along the direction normal to the dislocation line. By using a continuum theory, the so-called PN equation is used mathematically to describe the relation between the distribution of $S(x)$ [or $\rho(x)$] and a

[1]State Key Laboratory of New Ceramics and Fine Processing, School of Materials Science and Engineering, Tsinghua University, 100084 Beijing, China. [2]College of Mathematics and Physics, Beijing University of Chemical Technology, 100029 Beijing, China. [3]Institute of Welding and Surface Engineering Technology, Faculty of Materials and Manufacturing, Beijing University of Technology, 100124 Beijing, China. [4]These authors contributed equally: Yi Li, Xiangyang Liu. ✉e-mail: wancl@mail.tsinghua.edu.cn

restoring force $F[S(x)]$ originated from the atomic misfit:[19]

$$\frac{K}{2\pi}\int_{-\infty}^{+\infty}\frac{1}{x-x'}\frac{dS(x')}{dx'}dx' = \frac{K}{2\pi}\int_{-\infty}^{+\infty}\frac{\rho(x')}{x-x'}\cdot dx' = F[S(x)] \qquad (1)$$

where $K$ is the elastic coefficient as a function of dislocation character[19]. The equation can be solved by calculating the restoring force based on the generalized stacking fault (GSF) energies[19] which are estimated accurately by using the first principles. The continuous distribution of shear $S(x)$ combined with the GSF energies yields the energy barrier for the glide of dislocation, which is used to calculate the Peierls stress. There have been large amounts of dislocation research based on the PN model[20–31]. The GSF calculations apply a rigid model to address atomic structure, in which the atoms are only allowed to relax along with the directions normal to shear direction[20–31]. Accordingly, the GSF energy is purely inelastic strain energy. Although this approach has gained a great deal of success in metals and alloys[22–29], it is still highly questionable to be applied to non-metallic materials. The mixed feature of covalent and ionic bonding of non-metallic materials can bring some new challenges to correctly model the dislocation process.

This question has been noticed in some previous studies on the dislocation properties of materials beyond metals and alloys. For example, in strontium titanate $SrTiO_3$, a prototypical perovskite oxide with surprising plasticity[1–8], several studies on modeling the dislocation core of $SrTiO_3$ have been reported[20, 21, 32, 33], which contributes to the understanding of the dislocation properties. However, the Peierls stress of $SrTiO_3$ was underestimated by several orders of magnitude by using the GSF approach[20, 30], which suggests that the dislocations dynamics have not yet been fully understood. Meanwhile, we find the GSF approach cannot address imperfect structures because of the inability to consider the strain near the imperfection. We ascribe these problems to the fact that the traditional GSF approach limits the freedom degree of atoms, and accordingly, only inelastic strain energy is taken into account. However, in the PN model, the restoring force $F[S(x)]$ originates from the strain energies including both elastic and inelastic parts[18, 19]. The ignorance of the elastic part makes the GSF approach fail to account for the bond-breaking and re-bonding process in non-metallic materials, resulting in an inaccurate physical picture of the dislocation motions.

Some researchers have recognized the significance of elastic strain energy in modeling the dislocations of non-metallic materials. Carrez et al. applied a so-called Peierls–Nabarro–Galerkin (PNG) method[34], which yields a Peierls stress of 350 MPa for the ⟨110⟩ {110}-type screw dislocation of $SrTiO_3$. However, they estimated the elastic strain energy by using an element-free Galerkin method without explicit consideration of the atomic scale details of dislocation cores. Therefore, the physical mechanism of the slip process that is closely related to the atomic structure of $SrTiO_3$ cannot be provided. Guénolé et al. discussed the synchroshear dislocation in Laves phases by using the nudged elastic band (NEB) approach[35]. The NEB method is good at finding the slip path of dislocation but is not applicable to calculating Peierls's stress and the structure of the dislocation core.

To address the dislocation properties in non-metallic materials, we put forward a strategy to model the dislocation core and its motion by using local misfit energy instead of GSF energy. The in-plane freedom degrees are introduced for the atoms near the shear plane in our strategy, so the misfit region is no longer restricted to the shear plane, which is different from the definition of stack fault. The energy required by the shear process is named "local misfit energy" to be distinguished from GSF energy. As a result, the ignored elastic strain energy in the GSF approach can be taken into account to describe some unique physical processes of dislocation motion in oxides and intermetallic compounds. The method is applied to simulate the ⟨110⟩ {110} edge dislocation of $SrTiO_3$. The major characteristics of

dislocations in $SrTiO_3$, including the dislocation structure, average misfit energy, and Peierls stress, are all determined by the earlier stage of plastic deformation, in which the edge dislocations play the dominating role at room temperatures[3, 4, 6]. By contrast, the screw-type dislocation is hardly observed until the later stage of plastic deformation[6], so it is not involved in this work. The shear dependence of local misfit energy is analyzed in terms of bonding characteristics and electrostatic interactions, showing the difference from the results of the GSF approach[20, 21, 30]. The difference is ascribed to the amelioration of the simulation on the breaking and re-bonding process of the Ti–O bonds. The calculated Peierls stress $\sigma_{PN}$ is 305 MPa. An extrapolation of critical resolved shear stress is performed based on the experimental data[3], which agrees well with the calculated $\sigma_{PN}$.

The method in this paper also handles the interaction between dislocations and vacancies, which the traditional GSF approach fails to deal with. It is widely believed that point defect usually shows a pinning effect on dislocation and accordingly, reduces the dislocation mobility[36–41]. However, the calculation based on our model indicates that oxygen vacancies in $SrTiO_3$ can reduce the average misfit energy and the Peierls stress, contributing to the nucleation and activation of ⟨110⟩ {110} edge dislocation. These defects can help prevent crack extension, especially for ceramics in which crack propagation is easier than the nucleation of dislocations. The compression tests on $SrTiO_3$ single crystals directly evidence the effect of oxygen vacancies on reducing the critical shear stress required for activating the dislocations to glide. The micro-indentation measurements verify the contribution of oxygen vacancies in preventing crack extension, which can ameliorate the fracture toughness of materials. This paper provides a theoretical approach to model dislocation processes in non-metallic materials with complex chemical bonding. We believe it can bring new implications to ameliorate the intrinsic brittleness of non-metallic materials and explore compositions with abnormal ductility.

## Results and discussions
### Model for local misfit energy calculation
The PN model has been widely applied in metallic materials to analyze plasticity mechanisms, which has been summarized in several papers[19, 20]. In this paper, we introduced the in-plane degrees of freedom for the atoms near the glide plane to accurately describe the bond-breaking and re-bonding process of complex chemical bonds. Accordingly, the misfit region is no longer restricted to the glide planes, and the shear cannot be regarded as a stacking fault. Thus, we applied local misfit energy instead of the GSF energy to calculate the restoring force acting between the atoms on the two sides of the glide plane, which is required for solving the Peierls–Nabarro equation [Eq. (1)]. The local misfit energy ($\gamma$) is calculated by applying shear displacements [$S(x)$] to the atoms between the two glide planes. The values of $\gamma$ were defined as the variation of free energy normalized by the area of the shear plane. This definition has the same form as that of the GSF energy[19], but their computational models are significantly different.

Figure 1 shows the $SrTiO_3$ supercell, containing 120 atoms, used for the calculations on the local misfit energy of [011](0$\bar{1}$1) glide system. The atoms in the black regions are fixed and the other regions are allowed to fully relax in all directions when optimizing the supercell. The values of $S(x)$ are measured through the atoms of the two fixed regions instead of the atoms in the neighborhood of the glide planes. The relaxed regions are sandwiched between the fixed regions, forming a periodic structure. The sandwich structure is computationally demanding when compared with the slab structure in previous work[20]. However, the periodic structure does not include vacuum layers. It prevents the polarization caused by surface charges, which is of significance for calculating the static potentials in the supercell.

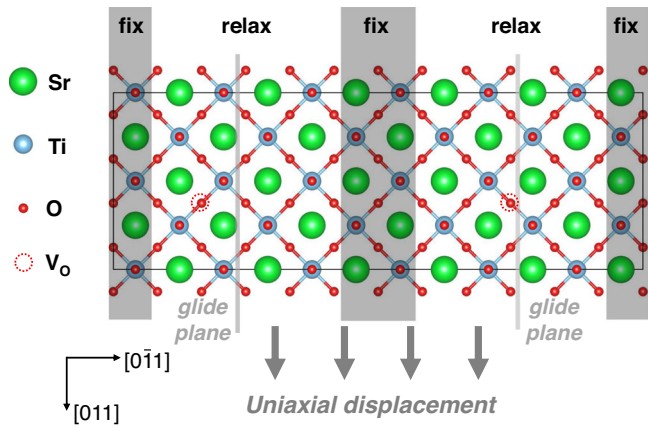

**Fig. 1 | Model for the calculations of local misfit energy.** The SrTiO₃ supercell contains 120 atoms and includes two glide planes, marked by pink lines, to maintain the periodic condition. All the atoms excluding those in the dark regions are allowed to fully relax in all directions. The un-filled dashed line circles mark the positions in which the oxygen vacancies $V_O$ are selected when calculating the deficient structure.

## [011](0$\bar{1}$1) edge dislocation in SrTiO₃

The local misfit energies $\gamma$ of [011](0$\bar{1}$1) system in SrTiO₃ are calculated using the model illustrated in the former section. The length of Burgers vector |**b**| is 5.5774 Å. The $S(x)$ dependence of $\gamma$ is distinguished from the previous results calculated by the GSF approach[20,21,30]. As shown in Fig. 2a, the $\gamma$ values begin to decrease when the shear displacement reaches 0.3|**b**|. Before further investigating the structure of the dislocation core and the Peierls plasticity, it is necessary to analyze the rationality of this abnormal $\gamma$–$S(x)$ dependence. The local misfit energies are calculated from the variation of free energy ($\Delta E_{\text{free}}$), which are summarized in Fig. 2b. The energy values of the initial structure are set as zero position. According to the DFT results, $\Delta E_{\text{free}}$ consists of three parts:

$$\Delta E_{\text{free}} = \Delta E_{\text{electrostatic}} + \Delta E_{\text{xc}} + \Delta E_{\text{eigen}} \qquad (2)$$

where $\Delta E_{\text{electrostatic}}$, $\Delta E_{\text{xc}}$, and $\Delta E_{\text{eigen}}$ are the variations of electrostatic energy, exchange-correlation energy, and Kohn–Sham eigenvalues, respectively[42]. The first part corresponds to the electrostatic interactions among ions and electrons. The last two parts are related to charge distributions and bonding characteristics. The $S(x)$ dependences of $\Delta E_{\text{electrostatic}}$ and $\Delta E_{\text{xc}} + \Delta E_{\text{eigen}}$ are shown in Fig. 2b. It is clear that the abnormal behavior occurs when $S(x)$ increases from 0.25|**b**| to 0.3 |**b**| for both $\Delta E_{\text{electrostatic}}$ and $\Delta E_{\text{xc}} + \Delta E_{\text{eigen}}$.

The charge densities of the SrTiO₃ supercell are applied to analyze the abnormal behavior of the sum of $\Delta E_{\text{xc}}$ and $\Delta E_{\text{eigen}}$. Figure 2c shows the two-dimensional (2D) displays of the charge densities on (100) Ti–O plane for the supercell with the shear displacements of 0.25|**b**|, 0.3|**b**|, 0.4|**b**| and 0.5|**b**|. Some Ti–O bonds break at the beginning of the shear process, which is marked by the black circles in the figure. Then the re-bonding of Ti–O, marked by the blue circles, is achieved through the lattice strain when the shear displacement reaches 0.3|**b**|. The break of Ti–O bonding lifts the values of $\Delta E_{\text{xc}} + \Delta E_{\text{eigen}}$, but the re-bonding process leads to a sudden drop of $\Delta E_{xc} + \Delta E_{\text{eigen}}$. It is widely believed that the Ti–O bonding is covalent and presents a strongly directional feature. The model in this paper enhances the freedom degree of the atoms near the glide planes. This makes it possible for the separated Ti and O atoms to re-bond together through the lattice strain instead of gradually getting close with the increasing shear displacements. The new Ti–O bonds are reinforced as the shear displacement increases, which further reduces the values of $\Delta E_{\text{xc}} + \Delta E_{\text{eigen}}$.

The electrostatic potentials of the SrTiO₃ supercell are calculated to analyze the $S(x)$ dependence of $\Delta E_{\text{electrostatic}}$. Figure 2d shows the planar averaged electrostatic potentials along **z**-direction, which corresponds to [0$\bar{1}$1], for the supercell with the shear displacements of 0, 0.15|**b**|, 0.25|**b**|, 0.3|**b**|, 0.4|**b**| and 0.5|**b**|. The plot for the initial position [$S(x) = 0$] has valleys and peaks. The valleys show two kinds of low and high, which correspond to the SrTiO and O₂ layers, respectively (Supplementary Fig. 1). The peaks appear in the middle of the SrTiO and O₂ layers. The shear initially makes the electrostatic potentials increase from the SrTiO layer to the O₂ layer and decrease from the O₂ layer to the SrTiO layer along [0$\bar{1}$1] direction. Generally, the total electrostatic potentials decrease until $S(x)$ reaches 0.25|**b**|. When the shear displacement reaches 0.3|**b**|, the electrostatic potentials symmetrically distribute on both sides of the glide planes, and high potential barriers are formed near the glide planes. The change is also presented in Supplementary Fig. 2, which shows the 2D displays of the electrostatic potentials on (100) Ti–O and Sr–O planes. The increase of $\Delta E_{\text{electrostatic}}$ stems from the high potential barriers.

The electrostatic potentials are closely related to the lattice polarization, the macroscopic electronic polarization of the SrTiO₃ supercell is calculated by using the Berry phase expressions to explore the origin of the potential barriers[43–46]. The total changes of dipole moment $\Delta p_z$, including both the ionic part and electronic part are shown in Fig. 2e. The dipole moment of the SrTiO₃ supercell without shear is set as zero point. The values of $\Delta p_z$ linearly increase with $S(x)$ but become almost zero after the shear displacement reaches 0.3|**b**|. It is widely recognized that the polarization in SrTiO₃ originates from the distortions of the TiO₆ octahedra. The local dipole moment is along the distortion direction of the TiO₆ octahedron. As shown in the insert figures of Fig. 2e, the TiO₆ octahedra on either side of the glide plane distort in the same direction for the case of $S(x) = 0.25$|**b**|. The increase in shear displacement strengthens the distortions of the TiO₆ octahedra. It contributes to the increase of $\Delta p_z$ when the distortion directions are the same. However, the distortion directions on one side of the glide plane are reversed after $S(x)$ reaches 0.3|**b**|. In this case, the total dipole moment always keeps zero because the local dipole moments on the two sides of the glide plane are aligned in the opposite direction.

Based on the calculations above, we propose a simple model as shown in Fig. 2f to explain the variation of electrostatic potentials in Fig. 2d. The SrTiO layers are alternately stacked with the O₂ layers along [0$\bar{1}$1] direction, constituting the lattice structure of SrTiO₃. The SrTiO and O₂ layers have positive and negative charges, respectively. Thus, the electric field (**E**) starts from the SrTiO layers and ends with the O₂ layers. The electrostatic potentials increase/decrease when the electric field (**E**) and the dipolar electric field (**P**) have the same/opposite directions. Before $S(x)$ reaches 0.3|**b**|, the dipolar electric field (**P**) always points to [0$\bar{1}$1] direction. Thus, the electrostatic potentials increase from the SrTiO layer to the O₂ layer and decrease from the O₂ layer to the SrTiO layer along [0$\bar{1}$1] direction. After $S(x)$ reaches 0.3|**b**|, the direction of **P** above the glide plane is reversed. As a result, all the local dipole moments point to the O₂ layer in the neighborhood of the glide plane, which raises the high potential barriers near the glide plane and contributes to the increase of $\Delta E_{\text{electrostatic}}$.

Generally, the behavior of $\gamma$ is the result of the re-bonding process and the reversal of dipoles. The properties of the [011](0$\bar{1}$1) edge dislocation were calculated based on the methodology proposed by Joos et al.[19]. Here we use the gradient of local misfit energy instead of the GSF energy to calculate the restoring force $F(S)$: $F(S) = -\partial\gamma/\partial S$. The local misfit energies are interpolated with the spline algorithm before the calculation. According to the PN equation [Eq. (1)], the restoring force

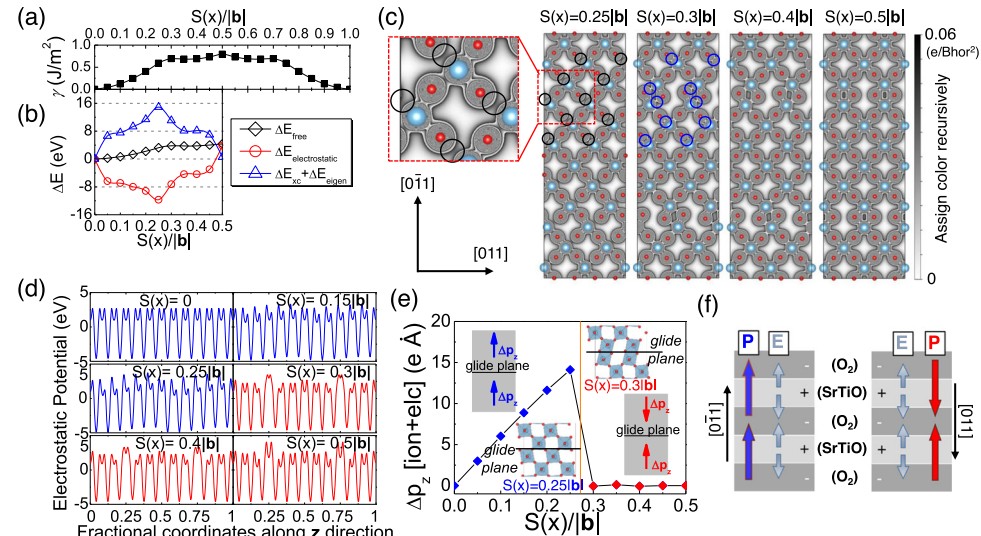

**Fig. 2 | Local misfit energy calculations for SrTiO₃. a** Local misfit energies γ and **b** the variations of free energy $\Delta E_{\text{free}}$, electrostatic energy $\Delta E_{\text{electrostatic}}$, the sum of exchange-correlation energy and Kohn–Sham eigenvalues $\Delta E_{\text{xc}} + \Delta E_{\text{eigen}}$ as a function of normalized shear displacement $S(x)/\mathbf{b}$ for [011]($0\bar{1}1$) system in SrTiO₃. **c** The two-dimensional displays of the charge densities on (100) Ti-O plane for the supercell with the shear displacements $S(x)$ of 0.25|**b**|, 0.3|**b**|, 0.4|**b**| and 0.5|**b**|. The circles mark the positions where the Ti-O bonds break [$S(x) = 0.25|\mathbf{b}|$] and recover [$S(x) = 0.3|\mathbf{b}|$]. **d** The planar averaged electrostatic potentials along **z** direction,

corresponding to [$0\bar{1}1$], for the supercell with the shear displacements $S(x)$ of 0, 0.15|**b**|, 0.25|**b**|, 0.3|**b**|, 0.4|**b**| and 0.5|**b**|. **e** Total changes of dipole moment $\Delta p_z$ including both ionic part and electronic part. The insert figures show the TiO₆ octahedra near the glide plane of the supercell with the shear displacements of 0.25|**b**| and 0.3|**b**|. **f** Schematic diagram of the model for explaining the variation behavior of electrostatic potentials. The blue/red colors in **d**–**f** correspond to the cases of parallel/anti-parallel local dipole moments. Source data are provided as a Source Data file.

$F[S(x)]$ are fitted through the equations[19]

$$S(x) = \frac{b}{2} + \frac{b}{\pi} \sum_{i=1}^{N} \alpha_i \cdot \arctan \frac{x - x_i}{c_i} \tag{3}$$

$$F(x) = \frac{Kb}{2\pi} \sum_{i=1}^{N} \alpha_i \cdot \frac{x - x_i}{(x - x_i)^2 + c_i^2} \tag{4}$$

where $\alpha_i$, $x_i$, and $c_i$ are fitting constants, $K$ is the elastic coefficient as a function of dislocation character[19], $b$ is the length of Burgers vector. The $\alpha_i$ are restricted by $\Sigma \alpha_i = 1$. $N$ is the fitting order with the value of $N = 9$ in this calculation. As shown in Fig. 3a, the fitting curve of the restoring force is in good accordance with the data calculated from the local misfit energies. The $x$-dependence of shear $S(x)$ calculated from the fitting curve is shown in Fig. 3b. The dislocation density $\rho(x)$ is the differential of the shear distribution, i.e., $\rho(x) = dS(x)/dx$. Three peaks of $\rho(x)$ are observed, which are different from the results of the GSF approach in the previous reports[20,21,30]. One peak corresponds to the center of the dislocation core, and the other two correspond to the positions where the re-bonding of Ti-O and the reversal of dipoles occur ($S(x) \approx 0.3|\mathbf{b}|$ and $0.7|\mathbf{b}|$).

High-resolution transmission electron microscopy (HRTEM) is applied to support our calculations on the core structure of [011]($0\bar{1}1$) edge dislocation. The HRTEM image as shown in Fig. 3d contains two partial ⟨011⟩ {011}-type dislocations in [100] oriented SrTiO₃ crystal. The long Burgers vector of [011]($0\bar{1}1$) system makes the ⟨011⟩ dislocations easy to dissociate into two parts with a stacking fault[21]. The HRTEM image is not clear near the right partial dislocation, so the area overlaps its Fourier-filtered image in order to show this part. The normalized shear distribution $S(x)/\mathbf{b}$ is only measured from the left partial dislocation (details in Supporting materials) because another part is too vague. The results are also included in Fig. 3b. According to the figure, the $x$ dependence of $S(x)/\mathbf{b}$ calculated from the PN model agrees well with the HRTEM results, which provides direct evidence for the validity of our simulation.

The plasticity of SrTiO₃ is further investigated from the Peierls stress. The total misfit energy $W(u)$ of the [011]($0\bar{1}1$) edge dislocation can be obtained by the sum of local misfit energy:[19]

$$W(u) = \sum_{m=-\infty}^{\infty} \gamma[S(ma - u)]a \tag{5}$$

where $a$ is the periodicity of $W(u)$. The results are shown in Fig. 3c. The Peierls stress is calculated by

$$\sigma_{\text{PN}} = \max\left[\frac{1}{b}\frac{dW(u)}{du}\right] \tag{6}$$

The calculated properties of the [011]($0\bar{1}1$) edge dislocation are summarized in Table 1. The elastic coefficient, the periodicity of misfit energy, the maximum restoring force, and the distribution range of the dislocation core in this work agree with the results of the GSF approach. However, the Peierls energy barrier ($\Delta W$) is two orders of magnitude larger than that of the GSF approach. Accordingly, the Peierls stress increases from 4 MPa (GSF model) to 305 MPa (our model). In order to compare the calculated $\sigma_{\text{PN}}$ with the experimentally measured critical resolved shear stress (CRSS), an extrapolation of CRSS has been performed down to 0 K based on the experimental data[3] (details in the Supporting materials). The extrapolated CRSS is 290 MPa at 0 K, which agrees well with the calculated Peierls stress in our manuscript (Supplementary Fig. 4). The optimization of the calculated Peierls stress can be ascribed to the improved degree of freedom for the atoms near the glide planes, which enables the lattice to stabilize by a strain during the shear process. The re-bonding of Ti-O and the reversal of dipoles take place through the lattice strain, leading to a dramatic variation of the misfit energy. The variation results in the three-peak form of $\rho(x)$ and promotes the Peierls energy barrier as well as the Peierls stress.

According to the traditional theory of the PN model, the Peierls stress is mainly determined by the parameters of $K$ and $\tau^{\text{max}}$[19]. The values of these two parameters in our calculations are close to the results in the GSF model as shown in Table 1, but the Peierls stress is

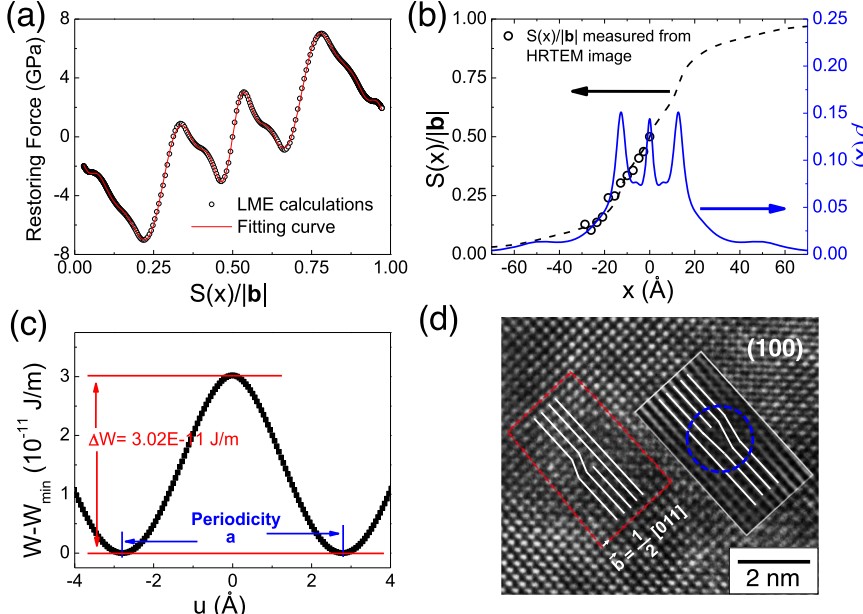

**Fig. 3 | Dislocation properties of SrTiO₃. a** Restoring force as a function of normalized shear $S(x)/|\mathbf{b}|$, where $|\mathbf{b}|$ is the length of Burger vector, 5.5774 Å; **b** normalized shear distribution $S(x)/|\mathbf{b}|$ and dislocation density $\rho(x)$ as a function of the distance $x$ from the dislocation core. It should be noted that $\rho(x)$ refers to the density of infinitesimal dislocations in Eq. (1). The unfilled circles in this figure denote the normalized shear distribution measured from the HRTEM image as shown in (**d**); **c** variation of the misfit energy $W$ as a function of the position $u$ of the dislocation core for the [011](0$\bar{1}$1) edge dislocation, the minimum misfit energy $W_{min}$ is 362.12E−11 J/m; **d** high-resolution transmission electron microscopy (HRTEM) image containing two partial ⟨011⟩ {011}-type dislocations from the [100] perspective. The Burgers circle yields a Burgers vector $\frac{1}{2}$[011] of the left partial dislocation. The HRTEM image near another partial dislocation is not clear, so the area, encircled by the rectangle, overlaps its Fourier-filtered image. The right partial dislocation is marked by the dotted line circle. Source data are provided as a Source Data file.

**Table. 1 | The elastic coefficient ($K$), the periodicity of total misfit energy ($a$), the maximum of restoring force ($\tau^{max}$), the half-width of dislocation core ($\zeta$), the Peierls energy barrier ($\Delta W$), the average misfit energy ($W_{ave} = \frac{1}{a}\int_0^a W(u)\mathrm{d}u$) and the Peierls stress ($\sigma_{PN}$) for the [011](0$\bar{1}$1) edge dislocation in SrTiO₃ calculated based on the PN model**

| | | $K$ (GPa) | $a$ (Å) | $\tau^{max}$ (GPa) | $\zeta$ (Å) | $\Delta W$ (10⁻¹¹ J/m) | $W_{ave}$ (10⁻¹¹ J/m) | $\sigma_{PN}$ (MPa) |
|---|---|---|---|---|---|---|---|---|
| $V_O$ free | Our model | 144.37 | 5.577 | 7.0 | 14.9 | 3.02 | 363.63 | 305 |
| | GSF model[20] | 144.07 | 5.571 | 6.7 | 15.5 | 0.04 | – | 4 |
| $V_O$ | Our model | 144.37 | 5.577 | 5.4 | 12.5 | 0.12 | 343.81 | 15 |

2$\zeta$ corresponds to the region wherein the shear $S(x)$ is larger than half of its maximum value.

two orders of magnitude larger than that of the GSF model. Besides, the average of the misfit energy $W_{ave}$ also deviates from the classical value $Kb^2/2\pi$[19]. These severe differences indicate that the traditional theory is no longer suitable for describing the dislocation core with multiple peaks. As mentioned above, the three-peak form stems from the re-bonding of Ti-O and the reversal of dipoles, which are the typical features of materials with strongly directional bonding. Further study on the PN model is required to physically describe the dislocation properties of this kind of material.

**Effects of oxygen vacancies on dislocation**

The calculations on the [011](0$\bar{1}$1) local misfit energy $\gamma$ of SrTiO₃ with $V_O$ also applied the model in Fig. 1. Two oxygen vacancies are introduced as shown in the figure. The length of Burgers vector $|\mathbf{b}|$ is 5.5774 Å. As shown in Fig. 4a, the $\gamma$ values calculated by the GSF approach are also set in Fig. 4a as a comparison. The minimum of $\gamma$ calculated by the GSF approach deviates from the initial position at which $S(x) = 0$. As a result, the restoring force is not continuous under periodic conditions. The singularity at the initial position indicates that the GSF approach fails to describe the dislocation properties of SrTiO₃ with $V_O$. Oxygen vacancies should lead to a lattice strain, which moves with the migration of $V_O$ in the shear process. However, the atoms are limited to relaxing only along the direction perpendicular to glide

planes in the GSF model. Thus, the system with $V_O$ cannot reach a stable state with the lattice strain, resulting in the singularity at the initial position.

As shown in Fig. 4a, the $\gamma$ minimum returns to the initial position when applying the model. As mentioned in the former section, our model removes the restrictions on the freedom degree of the atoms. Therefore, it is able to calculate the lattice strain extending to several atomic layers near $V_O$. The strain introduced by $V_O$ alters the direction of lattice distortion and makes the local lattice dipoles always point to the O₂ layers next to the glide planes. As a result, the electrostatic potentials of the oxygen-deficient structure symmetrically distribute on both sides of the glide planes, even though the shear displacement is smaller than 0.3$|\mathbf{b}|$ (Supplementary Fig. 5). In other words, the reversal of lattice dipoles is absent in the shear process of the oxygen-deficient structure. The local misfit energy is only governed by the breaking and re-bonding process of Ti-O bonds and the $\gamma$ values increase monotonously and smoothly before $S(x)$ reaches 0.5$|\mathbf{b}|$.

The properties of the [011](0$\bar{1}$1) edge dislocation of SrTiO₃ with $V_O$ are calculated via the same methods in the former section. As shown in Fig. 4b, the fitting curve of the restoring force is in good accordance with the data calculated from the local misfit energies. Figure 4c shows the $x$-dependence of $S(x)$ and $\rho(x)$. There is only one peak of $\rho(x)$ at the center of the dislocation core. The two peaks of

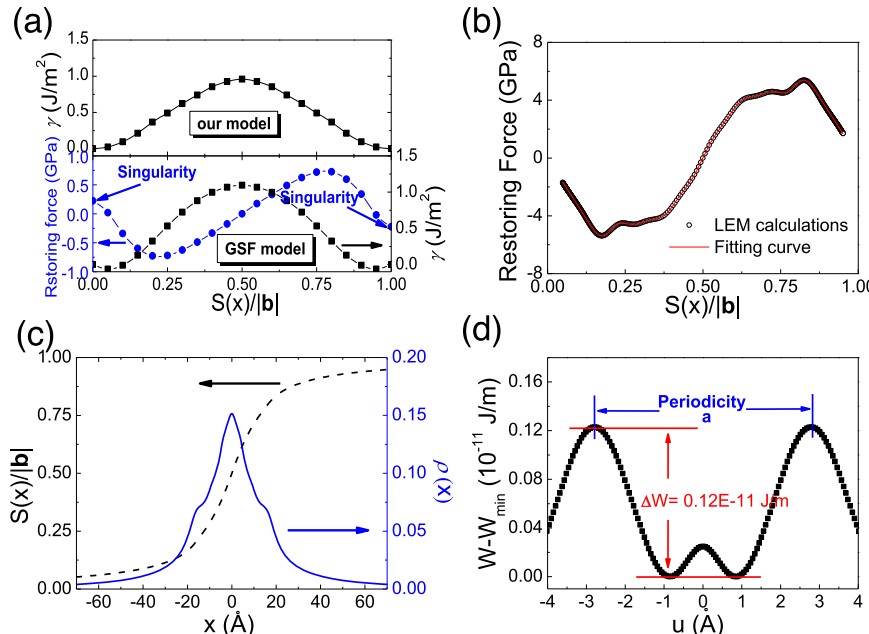

**Fig. 4 | Dislocation properties of SrTiO₃ with $V_O$. a** Local misfit energies $\gamma$ and GSF energies as a function of normalized shear $S(x)/|\mathbf{b}|$, where $|\mathbf{b}|$ is the length of Burger vector, 5.5774 Å. The restoring force calculated by GSF model is also shown in this figure; **b** Restoring force as a function of normalized shear $S(x)/|\mathbf{b}|$; **c** normalized shear distribution $S(x)/|\mathbf{b}|$ and dislocation density $\rho(x)$ as a function of the distance $x$ from the dislocation core. It should be noted that $\rho(x)$ refers to the density of infinitesimal dislocations in Eq. (1); **d** variation of the misfit energy $W$ as a function of the position $u$ of the dislocation core for the [011](0Ī1) edge dislocation of SrTiO₃ with $V_O$. $W_{min} = 343.74E{-}11$ J/m. Source data are provided as a Source Data file.

$\rho(x)$ at $S(x) = 0.3|\mathbf{b}|$ and $0.7|\mathbf{b}|$ are disappeared after $V_O$ is introduced. The change can be attributed to the monotonous variation of $\gamma$ at $S(x) = 0.3|\mathbf{b}|$ and $0.7|\mathbf{b}|$, which originates from the absence of the reversal of lattice dipoles. As a result, the Peierls energy barrier dramatically decreases to $0.12E{-}11$ J/m (Fig. 4d), and the Peierls stress is reduced to 15 MPa (Table 1). Besides, the average misfit energy $W_{ave}$ decreases from $363.63E{-}11$ to $343.81E{-}11$ J/m² (Table 1). It is noted that the quantitative magnitudes are meaningless for experimental work because of the deviation of the $V_O$ concentration.

Generally, the effects of oxygen vacancies on the dislocations in SrTiO₃ can be divided into several stages: nucleation, activation, motion, and multiplication. The misfit energy and the Peierls stress are the criteria for the nucleation and activation of a dislocation, respectively. In our calculation, the average misfit energy decreases after introducing oxygen vacancies, indicating that oxygen vacancies contribute to the nucleation of dislocation in SrTiO₃. The results agree well with the recently published experimental works on SrTiO₃[47, 48], in which the SrTiO₃ with higher vacancy concentration favors the dislocation nucleation.

The calculated Peierls stress decreases after introducing oxygen vacancies in SrTiO₃, which suggests oxygen vacancies will help to activate dislocations to move. In order to evidence this effect, the compression test on the {100} SrTiO₃ crystals before and after reduction treatment is carried out at room temperature. The critical resolved shear stress required by the activation of ⟨110⟩ {110}-type dislocations can be calculated by the measured yield stress times the Schmid factor (-0.5 for ⟨110⟩ {110}-type dislocations). As shown in Supplementary Fig. 6, the yield stress is about 120 MPa for the intrinsic SrTiO₃ crystals, similar to the previous research[3]. The yield stress decreases to 77 MPa after the oxygen reduction treatment, indicating that oxygen reduction can significantly lower the critical shear stress. It strongly supports our calculated result that oxygen vacancies contribute to the decrease of the Peierls stress of ⟨110⟩ {110}-type dislocations.

To elucidate the effect of oxygen vacancies on the motion of dislocations, molecular dynamics (MD) simulations were performed to investigate the interaction between oxygen vacancies and a moving dislocation (details in the supporting materials). It can be concluded from the MD simulations that a single oxygen vacancy may be beneficial to the kink formation (Supplementary Fig. 8e and f), but has little effect on the dislocation motion (Fig. 5). On the contrary, the cluster of oxygen vacancies has a pinning effect on dislocation motion (Fig. 5). In a word, the oxygen reduction is adverse to the dislocation motion, which also agrees with the recently published experimental works[47, 48].

The dynamic process of dislocation is closely related to crack propagation. Dislocations can change the stress intensity near a crack tip, which is known as the shielding or anti-shielding effect[49]. It depends on whether the dislocations are emitted from (shielding) or absorbed by (anti-shielding) the crack tip[50]. The dynamic observation via TEM has proved that the dislocations in SrTiO₃ are the shielding type[51]. A large number of dislocations is required to effectively achieve the shielding effect. However, crack propagation is easier than the nucleation and multiplication of dislocations in ceramics[52]. The dislocations are difficult to nucleate or multiply near crack tips before the cracks extend. Thus, the great mobility of ⟨110⟩ {110} type dislocations is unfortunately useless for increasing the fracture toughness of SrTiO₃. According to our calculations, the oxygen vacancies in SrTiO₃ can reduce the average misfit energy and the Peierls stress of ⟨110⟩ {110} dislocation, contributing to the nucleation and activation of dislocation. When the newly nucleated dislocations are activated to move, they can serve as the source of the dislocation multiplication and further improve the dislocation content. Thus, the oxygen vacancies are in favor of preventing crack extension.

The micro-indentations are performed on the reduced and unreduced SrTiO₃ crystals to prove the effect of oxygen vacancies. According to the literature reported by Fang et al.[53], the dominating dislocation activities include the dislocation nucleation, motion and multiplication in the micro-scale indentation tests. Fig. 6a and b show the indentation pictures of the SrTiO₃ crystal before and after the reduction process. The slip bands along [010] and [001] directions are observed near the indentations. According to the three-dimensional

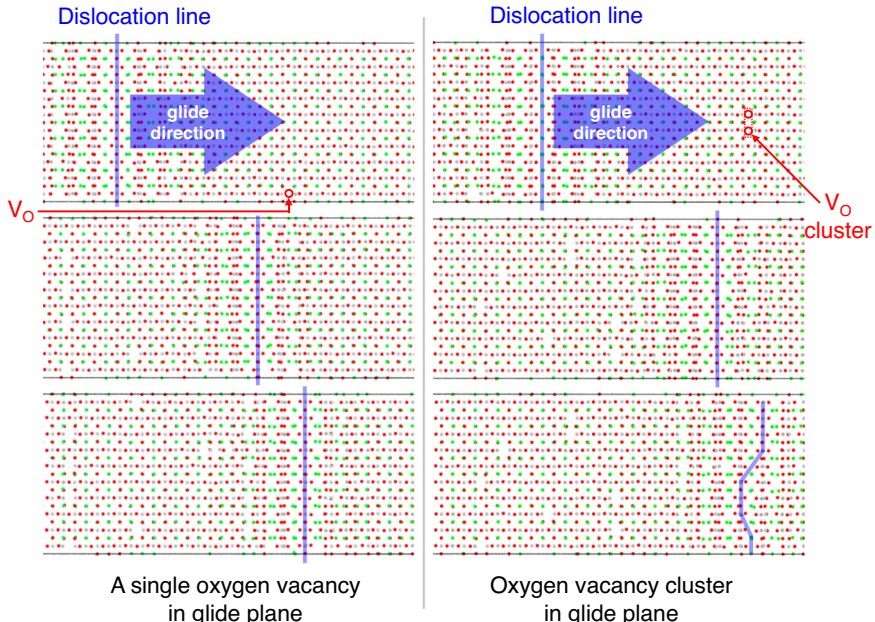

**Fig. 5 | Molecular dynamics simulations on [011]($0\bar{1}\bar{1}$) edge dislocation of SrTiO₃.** The migration of the dislocation line on the glide plane with a single oxygen vacancy (left figure) and a cluster of three oxygen vacancies (right figure). The simulation details are included in the supporting materials.

(3D) dislocation structure of SrTiO₃[54], the slip bands correspond to the glide of ⟨011⟩ {$0\bar{1}\bar{1}$} dislocations. The cracks in the unreduced sample extended beyond the area surrounded by the slip bands. The parts outside the surrounding area are marked by the red ellipses in Fig. 6a. The dislocations fail to shield the marked parts. However, after reduction, the slip bands spread to the ends of the cracks. It indicates that the shielding effect of dislocations covers the whole crack extension in the reduced sample. Clearly, the strong shielding effect shortens the crack length. As a consequence, the average crack length (*c*) decreases from 41.3 to 35.7 μm after introducing the oxygen vacancies (Fig. 6c). The results prove the above calculations on the ⟨011⟩ {$0\bar{1}\bar{1}$}-type edge dislocation. It should be noted that the dislocation multiplication is a dynamic process during the movement of dislocations, and its mechanism is usually complicated. The effect of oxygen vacancies on the dislocation multiplication in SrTiO₃ is still an open issue.

The hardness (*Hv*) and the fracture toughness measured in the indentation test are also shown in Fig. 6c. The hardness has little change after oxygen reduction treatment, which can be ascribed to two competing effects of both strengthening and weakening. A lower mobility of dislocation can reduce the size of the plastic zone produced by the indentation load, leading to a larger magnitude of hardness[55]. Oxygen vacancies can reduce the dislocation mobility of SrTiO₃ as mentioned above, and accordingly have a strengthening effect on hardness. Moreover, hardness generally shows a positive correlation to the elastic modulus[56]. It has been reported that the elastic moduli of ABO₃ perovskites usually decrease after oxygen reduction[57]. Thus, oxygen vacancies also have a weakening effect on the hardness of SrTiO₃. The competing effects of oxygen vacancies on hardness have also been reported in cerium oxide[58]. The fracture toughness is determined by the values of *Hv* and *c* [Eq. (8)]. After the oxygen reduction treatment, the hardness keeps almost unchanged while the crack length obviously decreases, leading to a 30% improvement of fracture toughness.

In conclusion, a strategy is proposed for investigating the dislocation properties within the framework of the Peierls–Nabarro model. The atoms near the shear plane are fully relaxed to calculate the local misfit energy instead of the generalized stacking fault. This enables the lattice to stabilize by a strain during the shear process, which is the key distinction of the method. The ⟨110⟩ {110} local misfit energies of SrTiO₃ are calculated based on the first principles. The variation trend of local misfit energies is different from that in the traditional GSF approach. The change is ascribed to the re-bonding of Ti–O and the reversal of localized dipoles, both of which are achieved by the lattice strain. The new trend of local misfit energies leads to a significant promotion of the Peierls barrier. Accordingly, the Peierls stress is two orders of magnitude larger than that calculated by the conventional GSF approach. The value of Peierls stress is 305 MPa, which agrees well with the extrapolated value from the reported experiment values.

Besides, the method also handles the interaction between the dislocation and oxygen vacancy in SrTiO₃, which the GSF approach fails to deal with. The amelioration is also ascribed to the high freedom degree of the atoms, which allows the lattice strain to extend to several atomic layers near oxygen vacancies. The results calculated from the Peierls–Nabarro model indicate that oxygen vacancies can reduce the average misfit energy and the Peierls stress of ⟨110⟩ {110} edge dislocation, which contribute to the nucleation and activation of dislocation. This conclusion is supported by the compression tests and the micro-indentation measurements on the {100} SrTiO₃ single crystals before and after oxygen reduction treatment. Apparently, A lower yield stress is observed in the oxygen-reduced specimen, which proves that oxygen vacancies contribute to decreasing the Peierls stress of ⟨110⟩ {110}-type dislocations. In the indentation test, the crack length decreased after oxygen reduction treatment, verifying the contribution of oxygen vacancies on the nucleation of dislocation. More importantly, The Vickers hardness remained almost unchanged, so the indentation fracture toughness of SrTiO₃ crystal can be improved by 30% through a simple reduction process.

This work provides a theoretical approach to model the dislocation process in non-metallic materials. The underlying mechanisms are disclosed, including the breaking and re-bonding processes of chemical bonds and the lattice dipole inversion. A good agreement between the calculated Peierls stress and experimental values is obtained, suggesting the validity of the model. We believe the model can provide an in-depth understanding of the dislocation processes in

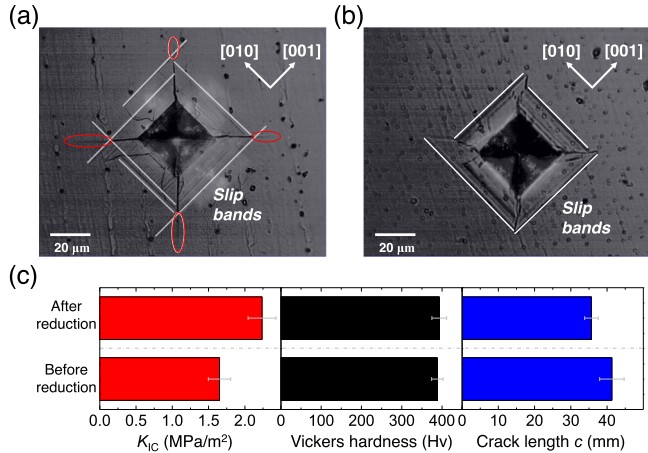

**Fig. 6 | Results of indentation tests on {100} SrTiO₃ crystals before and after oxygen reduction.** Indentation pictures of the SrTiO₃ crystal **a** before and **b** after the reduction process; **c** indentation fracture toughness $K_{IC}$, Vickers hardness $H_V$ and crack length $c$ measured in the indentation test (the error bars represent the standard deviations). Source data are provided as a Source Data file.

non-metallic materials, which can help tune the plasticity and explore ductile compositions.

## Methods
### First-principles calculations
The local misfit energies $\gamma$ of SrTiO₃ were calculated with Vienna ab initio simulation package (VASP)[59] on the basis of density-functional theory (DFT). Density-functional calculations were performed using the projector-augmented wave (PAW) method[60, 61]. Generalized gradient approximation (GGA) was employed as the electronic exchange-correlation potential. The plane wave cutoff energy was set as 500 eV. The supercell in Fig. 1 was built based on an optimized unit cell, whose cell parameter is 3.944 Å. A 3 × 1 × 1 Monckhorst−Pack $k$-mesh was used for the supercell to fully optimize the atomic positions. The convergence criteria are within 0.0001 meV for total energy and 5 meV/Å for Hellmann−Feynman force. The energy and polarization properties are calculated using a 5 × 3 × 1 Monckhorst−Pack $k$-mesh based on the optimized supercell.

### Experimental details
The SrTiO₃ {100} single crystals (Hefei CPI Co., Ltd) were used as initial materials. The oxygen reduction treatment was performed at 1450 °C in pure Argon atmosphere for 6 h. Oxygen vacancies and electrons are introduced simultaneously ($O_O^{\times} \rightarrow V_O^{\bullet\bullet} + 2e' + 1/2O_2$) during the reduction treatment, and they can contribute to the improvement of electrical conductivity. The impedance spectroscopy measurements are applied to compare the electrical conductivity of the reference sample and the reduced sample. The electrical conductivity of SrTiO₃ crystal increases from 8.22E−8 to 0.77 S/cm after the reduction treatment, indicating a high oxygen vacancy concentration in the reduced sample.

High-resolution transmission electron microscopy (HRTEM; JEOL, JEM-2100F, Japan) was used to map the [011](0$\bar{1}$1) edge dislocation core of SrTiO₃ crystal. Several slices were cut from raw crystals and polished to a thickness of 60−80 μm. The polished slices were thinned by an ion milling instrument (GATAN, G691, America) to satisfy the requirement of the HRTEM test.

A micro-indentation instrument (Omnimet MHT, Buehler, USA) was applied to observe the dislocation behavior of SrTiO₃ from the crack extension process. The raw crystals were polished and annealed at 900 °C in the air for 1 h before the indentation measurements. After the measurements, the samples were reduced to a

pure Argon atmosphere, and indentation tests were applied again. The fracture toughness ($K_{IC}$) and the Vickers hardness ($H_V$) were obtained from the equations:[62, 63]

$$K_{IC} = 0.018 \sqrt{\frac{E}{H_V}} \frac{P}{c^{1.5}} \qquad (7)$$

$$H_V = 0.464 \frac{P}{d^2} \qquad (8)$$

where $E$ is the elastic modulus obtained by an acoustic velocity test, $H_V$ is the Vickers hardness, $P$ is the load and was set as 0.3 kgf in the measurements, $c$ is the length from indentation center to crack tip, and $d$ is the half-length of indentation diagonal. 22 indentations were performed to calculate the averaged values of $K_{IC}$, $H_V$, and $c$.

The uniaxial compression tests were performed by using the SrTiO₃ crystals in the shape of quadrangular prisms with dimensions 2 × 2 × 4 mm³. The details on the specimen preparation are available in the supporting materials. The specimen surfaces were parallel to the {100} planes. The ⟨100⟩ compression axis was parallel to the longer side of the prisms. The compression test was performed by a universal testing machine (SHIMADZU, AG-IC, Japan) with a constant strain rate of $2.08 \times 10^{-4}$ s$^{-1}$ at room temperature.

## Data availability
The data that support the findings of this study are available from corresponding author upon request. Source data are provided with this paper.

## Code availability
The code that supports the findings of this study are available from corresponding author upon request.

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

## Acknowledgements

This work was supported by the China Key National R&D Plan (No. 2017YFA0700705), the National Natural Science Foundation of China (No. 52022042) and National Science and Technology Major Project (J2019-VII–0008-0148.

## Author contributions

Y.L. designed the research based on the discussion with C.W., Y.L. performed the first-principles calculations, Y.L. and X.L. performed the molecular dynamics simulations, Y.L., X.L., Y.H. and M.H. conducted the experiments including the characterization and the measurements, Y.L., X.L., P.Z. and C.W. did the analysis and wrote the manuscript. All of the authors contributed to manuscript preparation.

## Competing interests

The authors declare no competing interests.
