## [Peer Review File · Nature Communications]

Theoretical insights into the Peierls plasticity in SrTiO₃ ceramics via dislocation remodellingReviewers' comments:

Reviewer #1 (Remarks to the Author):

The manuscript "Theoretical insights into the Peierls plasticity in SrTiO₃ ceramics via dislocation remodelling" by Y. Li et al. reports a modified Peierls-Nabarro model by employing a local misfit energy (gradient) in the model, enabling a high accuracy prediction of the dislocation mobility, such as the Peierls stress, in a model system of SrTiO₃, representing the more difficult to model non-metallic material systems using the traditional method. Further, this methodology was applied to the oxygen deficient SrTiO₃ and found improved ductility, thus offering new implications to tune the plasticity and explore new ductile compositions.

It is generally believed that adding a local energy gradient term should improve the traditional generalized stacking fault (GSF) based on Peierls-Nabarro model. So the authors' exploration in this regard is certainly valuable. The question usually falls into how effective for any modified approach could achieve in terms of the desired accuracy? The comparison resulting in a close agreement between the calculations in the present manuscript and the experimentally measured range at finite temperatures is still not enough, to warrant its publication in Nature journals such as Nature Communication, in my view. Nowadays, a direct DFT calculation of the Peierls stress in non-metallic materials is computational workable, in combination with anisotropy elasticity theory to take care of the finite supercell size correction in DFT results. I suggest the authors to work on these calculations, in order to validate the effectiveness of the modified theory. If successful, the modified theory is envisioned to have great impact, for example, paying a way for high prediction power in high-throughput screening of non-metallic materials using the much faster Peierls-Nabarro model, as well as in material design.

Reviewer #2 (Remarks to the Author):

The authors present a study on an improved model to calculation of the Peierls stress for $\langle 110 \rangle \{110\}$ type dislocations in SrTiO₃ which should better account for the complex bond nature and the breaking and re-bonding process. The authors also attempt to relate their theoretical findings to experimentation. Progress in better modeling and a broader understanding around the Peierls stress in ceramics is timely needed and the topic merits interest from nature communications. Unfortunately, point 1 and 2 below may be very impactful concerns to the technical accuracy. Following those two key concerns, I suggest to reject the manuscript in its current form but to consider a re-submission. After that the list of smaller comments may be redone or extended. Nevertheless, I hope the comments to be helpful for the authors.

A series of questions towards the current version of the manuscript arises:

1) Line 299-301 "..., resulting in the singularity at the initial position." This seems strange. When consulting with Figure 4a, the restoring force (blue) is zero at the minimum of the energy landscape which should be okay. This rises the question about the burgers vector. In the rigid model, the spacing between the energy minima is about 5.5 Å. This makes sense as this is $\sqrt{2} \cdot 3.905 = b$. However, in the new model the energy minima are space 6.1 Å apart which is larger than the burgers vector of SrTiO₃. This is an unacceptable discrepancy and may be the reason for the unphysical mismatch. At the moment I do not see a possibility how a model can generate correct values with an incorrect burgers vector. Likely larger revision or detailed discussion is needed here.

2) Line 108/109 starts one of the key thoughts which, however, has a problem. The authors suggest that oxygen vacancies help dislocations to glide which is then supposed to reduce cracking and increase toughness. At first sight the experimental results appear to verify the simulation as shorter cracks are observed in the presence of more oxygen vacancies but there are several intermediate steps which need to be carefully discriminated.

I suggest to look at the mechanisms at two levels: Dislocation motion and the effect of dislocation on

toughness.

Effect of dislocations on toughness. As Rice derives in his 1992 manuscript (Dislocation Nucleation from a Crack Tip - an Analysis Based on the Peierls Concept) dislocations are easy to nucleate at crack tips in metals but for ceramics (at least at low temperature and high crack velocities), crack propagation is easier than the nucleation of dislocations. Therefore, even the types of ceramics which are ductile (as single crystals) at room temperature show, nevertheless, a brittle behavior. This means they are ductile and brittle at the same time requiring a careful discrimination of toughness and ductility. For the example of SrTiO₃, nucleation of dislocations at the crack tip does not occur and, hence, the great mobility of <110>{110} type dislocations is unfortunately useless for increasing the toughness. This is because dislocations are needed very close to the crack tip. But the natural density is much too low and nucleation from the tip does not occur which makes the nucleation the bigger problem over the mobility. Therefore, an effect on toughness can only be achieved if an extreme dislocation density is engineered. This is explained in detail in a recent publication: Porz L, et al. Dislocation-toughened ceramics. Mater Horiz. 2021;81528-37.

Secondly, the effect of vacancies on the dislocation motion has also several stages. Starting from small to large: If a vacancy sits in the core, it may alter the energy landscape and reduce effective Peierls barriers. At a slightly larger scale, it may help with the formation of kinks, which merely help overcome the Peierls barrier (very effectively) but do not alter the barrier itself. Detailed descriptions can be found here: Messerschmidt U. Dislocation dynamics during plastic deformation. Springer. (2010). So figure 5f suggests the vacancies effects the kink formation but this is a different effect than the effect on the Peierls barrier itself? What is the effect of the reduction treatment on the yield stress?

At the next level, vacancies may help reducing the stress needed for nucleation of new dislocations and lastly, they may be obstacles in the way of dislocations reducing their mobility in the end. A detailed study on the last two aspects was recently published: Stich S, et al. Room-temperature dislocation plasticity in SrTiO₃ tuned by defect chemistry. J Am Ceram Soc. 2021;1051318-29. The findings presented there actually suggest that, in the end, oxygen vacancies reduce the propagation of dislocations on a macroscopic scale which is opposite to the effect suggested here. Moreover, an altered mobility does not need to impact the toughness directly as elaborated above. Lastly, results from Vickers indentation needs to be used cautiously as reviewed by Quinn and Bradt in 2007 ("On the Vickers Indentation Fracture Toughness Test")

In summary, the highly desired correlation to experiment giving this study broader impact cannot be soundly done in the way presented.

3) Please give the Peierls-Nabarro equation for reference.

4) Figure 6 needs a scale bar.

5) Figure 1: The arrows drawn indicate a uniaxial stress but not a shear stress, please correct.

6) Disregistry. What does this word mean?

7) Figure 3b has the unit dislocation density in the surrounding of a dislocation on the y-axis.

Dislocation density quantifies the number of dislocations in a volume/area. The unit does not appear to make sense here.

8) While language is generally okay, there are multiple incidents where language is not at 100%. E.g. line 317 says "Fig. 4c shows the distribution of dislocation core." This seems incomplete or grammatically incorrect. A careful revision of the language (maybe with editorial assistance) may remedy this. Also, small things such as line 454 "data availability" not "date availability".

9) Line 101: experimental values are cited with ref. 18 (Nabarro 1947). Please provide the experimental value and try to find a reference specific to SrTiO₃ so that your derived value can be better compared.

10) Line 247. How can only half a b-vector be observed? Is this a statistically representative case? E.g. Jin et al. (TEM study of < 110 >-type 35.26 degrees dislocations specially induced by polishing of SrTiO₃ single crystals) and others found that partial dislocations only dissociated about 5 unit cells.

Reviewer #3 (Remarks to the Author):

The submitted manuscript deals with the estimation of the lattice resistance to dislocation motion in SrTiO₃ with and without the presence of an oxygen vacancy. The authors propose a new way of calculating the Peierls stress which increases the previous estimates from very few to hundreds of MPa, with the latter much closer to experimental observations.

Overall, I agree with the authors that this is an interesting and still very poorly understood topic and that new approaches that take into account the peculiarities of ordered crystals are needed.

However, I do have several reservations about the presented hypotheses, data from the literature and mechanisms:

- the main assumption, i.e. that in the literature the Peierls stress is solely estimated based on a GSF method that does not take into account any relaxation normal to the slip plane, is not a complete view of the literature. For example, Cordier, Carrez et al. have calculated the Peierls stresses of several complex crystals including perovskites and using GSFs on different crystal planes can achieve a good estimate of the core structure and Peierls stress. Among their work there is also an estimate of 350 MPa for the Peierls stress of SrTiO₃ (although they did estimate much much less before using a 1D estimate), which is very close to what the authors suggest here. An example of this work can be found here: <https://doi.org/10.1016/j.scriptamat.2010.04.045>

- there are also other methods which allow the estimation of energy barriers in deformation of complex crystals and which account for the fact that dislocation motion may not be planar or happen along a rigid shear direction. This is the case for example for synchroshear, which was first suggested for Al₂O₃ but is mainly associated with Laves phases. An example may be found here: <https://www.sciencedirect.com/science/article/pii/S1359646219301599>

- the authors plot in figure 5 and discuss in the text the acceleration of kink formation in the presence of a vacancy. However, there is no discussion as to what happens as the dislocation line travels beyond the vacancy - pinning is also an option, which may result in the opposite effect overall and I would suggest to discuss this explicitly, hoping that the available data allows this directly already.

- it is not quite clear why the authors relate dislocation mobility primarily to the more indirect measure of crack propagation, although slip bands are clearly indicated in their experimental figures. Is this as the surface obscures slip bands more completely after reduction? Why discuss the measured hardness directly. there is quite a bit of work on SrTiO₃, its dislocations and also the role of vacancies in these works that carry out a very careful dislocation etchpit analysis:

<https://doi.org/10.1016/j.scriptamat.2020.07.033> & <https://doi.org/10.1111/jace.18118>

Overall, I am therefore doubtful of the completeness of the work in terms of its comparison with other methods and how it is embedded in the literature, as some very basic claims made in the paper do not appear entirely accurate.

Response Letter

We appreciate the constructive and helpful suggestions from the reviewers and we have revised the manuscript accordingly. The changes are marked in the revised manuscript, and the responses to the comments are listed as follows:

Reviewer #1 (Remarks to the Author)

Comment_0: The manuscript “Theoretical insights into the Peierls plasticity in SrTiO₃ ceramics via dislocation remodelling” by Y. Li et al. reports a modified Peierls-Nabarro model by employing a local misfit energy (gradient) in the model, enabling a high accuracy prediction of the dislocation mobility, such as the Peierls stress, in a model system of SrTiO₃, representing the more difficult to model non-metallic material systems using the traditional method. Further, this methodology was applied to the oxygen deficient SrTiO₃ and found improved ductility, thus offering new implications to tune the plasticity and explore new ductile compositions.

Response_0: Thanks for the comments.

Comment_1: It is generally believed that adding a local energy gradient term should improve the traditional generalized stacking fault (GSF) based on Peierls-Nabarro model. So the authors’ exploration in this regard is certainly valuable. The question usually falls into how effective for any modified approach could achieve in terms of the desired accuracy? The comparison resulting in a close agreement between the calculations in the present manuscript and the experimentally measured range at finite temperatures is still not enough, to warrant it’s publication in Nature journals such as Nature Communication, in my view. Nowadays, a direct DFT calculation of the Peierls stress in non-metallic materials is computational workable, in combination with anisotropy elasticity theory to take care of the finite supercell size correction in DFT results. I suggest the authors to work on these calculations, in order to validate the effectiveness of the modified theory. If successful, the modified theory is envisioned to

have great impact, for example, paying a way for high prediction power in high-throughput screening of non-metallic materials using the much faster Peierls-Nabarro model, as well as in material design.

Response_1: Thanks for the comment and suggestion. We attempted to perform a direct DFT calculation of the Peierls stress as the reviewer suggested. According to our calculations in the manuscript, the size of $\langle 110 \rangle \{110\}$ edge dislocation in SrTiO_3 is about 5 nm. We built a $10 \times 11 \times 24$ supercell, in which two symmetrical dislocations are introduced using the Molecular Dynamics (MD) simulations following the same method as illustrated in the supporting materials. The process of building the dislocations is shown in the figure below.

Fig. R1 Schematic diagram for building the $[011](0\bar{1}1)$ edge dislocations.

For the purpose of minimizing the amount of calculation, the relaxed structure was cut into a $1 \times 11 \times 24$ supercell containing 2525 atoms. We used 15 nodes (64 cores per node) to carry on the DFT simulations on this large model, but the calculation was still difficult to run smoothly. First principles pseudopotential calculations are really computational demanding. In fact, the DFT calculations on the model with more than a thousand of atoms have not been reported by far. The VASP package required about 10 hours to handle a single ionic relaxation step for the dislocation model even though 960 cores were applied. Besides, technical errors were often reported because the supercell was too large for the electronic minimization algorithm in the VASP package. The current scale of computing resource is large enough for the parallel operation. Further increase of nodes has little effect on improving the computing efficiency because of the consumption between the nodes. Thus, unfortunately, it is still early to say the current DFT calculations are able to handle a dislocation model even with the minimum size.

Fig. S4 Extrapolation of CRSS calculated by fitting the experimental data corresponding to $\langle 110 \rangle \{ 110 \}$ edge dislocation [D. Brunner, Acta Mater. 2006, 54: 4999-5011].

As the reviewer mentioned, the comparison between the calculated Peierls stress σ_{PN} and the experimentally measured critical resolved shear stress (CRSS) measured at finite temperatures is not accurate enough. The minimum measured temperature is about 50 Kelvin in the reported experiments [P. Gumbsch et al., Phys. Rev. Lett. 2001, 87; D. Brunner, Acta Mater. 2006, 54: 4999-5011]. The issue has been addressed by performing an extrapolation of experimental data to 0 Kelvin [P. Carrez et al. Scripta Mater., 2010, 63: 434-437]. The influence of temperature on CRSS can be analyzed in terms of dislocation motion governed by the thermally activated process [P. Carrez et al. Scripta Mater., 2010, 63: 434-437; D. Brunner, Acta Mater. 2006, 54: 4999-5011]:

$$\Delta H^*(\sigma) = \Delta H_0 [1 - (\sigma / \sigma_{PN})^p]^q$$

where ΔH^* is the critical enthalpy of the configuration as a function of the effective stress σ , p and q are the fitting parameters. The values of $\Delta H_0 = 2.4$ eV, $p = 0.5$ and $q = 2.5$ are applied according to P. Carrez's work. $\Delta H^*(\sigma)$ can be converted into a function of critical resolved shear stress at constant strain rate and follows an Arrhenius-type relationship: $\Delta H^*(\sigma) = C k_B T$, where k_B is the Boltzmann constant, T is the absolute temperature and C is a factor that depends on σ and plastic strain rate. In the standard experimental conditions, C is usually in the range of 20-30 [M. Tang et al. Acta Mater.,

1998, 46: 3221; D. Brunner et al. Phys. Stat. Sol. (a), 1991, 124: 155], and a value of 20 is applied. An extrapolation of CRSS has been performed to 0 Kelvin based on the experimental data [D. Brunner, Acta Mater. 2006, 54: 4999-5011]. As shown in Fig. S4, the extrapolated CRSS is 298 MPa at 0 Kelvin, which agrees well with the calculated Peierls stress value (305 MPa) in our manuscript.

Besides the accuracy in the prediction of Peierls stress, our model also provides physical insights into the Peierls plasticity of SrTiO₃, such as the re-bonding process of the Ti-O bonds and the reversal of lattice dipoles, which has not been disclosed before. In addition, our theory provides a practical approach to deal with the complex model accompanied with defects, which the traditional GSF method fails to handle.

Revision_1: The relative discussions have been added in page 6, paragraph 1 and page 14, paragraph 1 of article and page 4 of the supporting materials.

Reviewer #2 (Remarks to the Author)

The authors present a study on an improved model to calculation of the Peierls stress for $\langle 110 \rangle \{110\}$ type dislocations in SrTiO₃ which should better account for the complex bond nature and the breaking and re-bonding process. The authors also attempt to relate their theoretical findings to experimentation. Progress in better modeling and a broader understanding around the Peierls stress in ceramics is timely needed and the topic merits interest from nature communications.

Unfortunately, point 1 and 2 below may be very impactful concerns to the technical accuracy. Following those two key concerns, I suggest to reject the manuscript in its current form but to consider a re-submission. After that the list of smaller comments may be redone or extended.

Response_0: Thanks for the encouraging evaluation and the valuable comments. Point 1 is a misread for the horizontal axis of Fig. 4a. For point 2, we have renewed our view on the dynamic process of dislocation based on the reviewers' constructive suggestions. New simulations and experiments were performed and the results were added to support our analysis. All the smaller comments have also been addressed.

Comment_1: Line 299-301 “..., resulting in the singularity at the initial position.” This seems strange. When consulting with Figure 4a, the restoring force (blue) is zero at the minimum of the energy landscape which should be okay. This rises the question about the burgers vector. In the rigid model, the spacing between the energy minima is about 5.5 Å. This makes sense as this is $\sqrt{2} \cdot 3.905 = b$. However, in the new model the energy minima are space 6.1 Å apart which is larger than the burgers vector of SrTiO₃. This is an unacceptable discrepancy and may be the reason for the unphysical mismatch. At the moment I do not see a possibility how a model can generate correct values with an incorrect burgers vector. Likely larger revision or detailed discussion is needed here.

Response_1: Thanks for the comment. It is actually a misunderstanding for Figure 4a. The magnitudes of horizontal axis were misread. As shown in the figure below, we used 5.5774 Å as the length of Burgers vector for both the rigid model and the new model. In the rigid model, the spacing between the energy minima is 5.0197 Å. The value is smaller than the length of Burgers vector, which is the origin of the singularity of the restoring force as we discussed in the manuscript.

Fig. 4(a) Local misfit energies γ and GSF energies as a function of shear $S(x)$.

Revision_1: In order to avoid this misreading, we normalized the horizontal axis by the Burgers vector b . Figure 4a was replaced by the new figure as shown below. The description “ $|b|$ is the length of Burger vector, 5.5774 Å” was also added in the figure caption.

Fig. 4(a) Local misfit energies γ and GSF energies as a function of normalized shear $S(x)/|b|$.

Comment_2: Line 108/109 starts one of the key thoughts which, however, has a problem. The authors suggest that oxygen vacancies help dislocations to glide which is then supposed to reduce cracking and increase toughness. At first sight the experimental results appear to verify the simulation as shorter cracks are observed in the presence of more oxygen vacancies but there are several intermediate steps which need to be carefully discriminated.

I suggest to look at the mechanisms at two levels: Dislocation motion and the effect of dislocation on toughness.

Effect of dislocations on toughness. As Rice derives in his 1992 manuscript (Dislocation Nucleation from a Crack Tip - an Analysis Based on the Peierls Concept) dislocations are easy to nucleate at crack tips in metals but for ceramics (at least at low temperature and high crack velocities), crack propagation is easier than the nucleation of dislocations. Therefore, even the types of ceramics which are ductile (as single crystals) at room temperature show, nevertheless, a brittle behavior. This means they are ductile and brittle at the same time requiring a careful discrimination of toughness and ductility. For the example of SrTiO₃, nucleation of dislocations at the crack tip does not occur and, hence, the great mobility of $\langle 110 \rangle \{110\}$ type dislocations is unfortunately useless for increasing the toughness. This is because dislocations are needed very close to the crack tip. But the natural density is much too low and nucleation from the tip does not occur which makes the nucleation the bigger problem over the mobility. Therefore, an effect on toughness can only be achieved if an extreme

dislocation density is engineered. This is explained in detail in a recent publication: Porz L, et al. Dislocation-toughened ceramics. *Mater Horiz.* 2021;81528-37.

Secondly, the effect of vacancies on the dislocation motion has also several stages. Starting from small to large: If a vacancy sits in the core, it may alter the energy landscape and reduce effective Peierls barriers. At a slightly larger scale, it may help with the formation of kinks, which merely help overcome the Peierls barrier (very effectively) but do not alter the barrier itself. Detailed descriptions can be found here: Messerschmidt U. *Dislocation dynamics during plastic deformation.* Springer. (2010). So figure 5f suggests the vacancies affect the kink formation but this is a different effect than the effect on the Peierls barrier itself? What is the effect of the reduction treatment on the yield stress?

At the next level, vacancies may help reducing the stress needed for nucleation of new dislocations and lastly, they may be obstacles in the way of dislocations reducing their mobility in the end. A detailed study on the last two aspects was recently published: Stich S, et al. Room-temperature dislocation plasticity in SrTiO₃ tuned by defect chemistry. *J Am Ceram Soc.* 2021;1051318-29. The findings presented there actually suggest that, in the end, oxygen vacancies reduce the propagation of dislocations on a macroscopic scale which is opposite to the effect suggested here. Moreover, an altered mobility does not need to impact the toughness directly as elaborated above. Lastly, results from Vickers indentation need to be used cautiously as reviewed by Quinn and Bradt in 2007 (“On the Vickers Indentation Fracture Toughness Test”).

Response_2: Thanks for the reviewer’s detailed suggestions, which really help us correct our opinions on the dynamic process of dislocation. The process can be divided into three stages: nucleation, activation and motion of dislocations. The misfit energy and the Peierls stress are the criteria for nucleating and activating a dislocation. It is improper to use them to analyze dislocation motion as we did. In our calculation, the average misfit energy [$W_{ave} = \frac{1}{a'} \int_0^{a'} W(u) du$] and the Peierls stress of the dislocation decrease after introducing oxygen vacancies in SrTiO₃. The results indicate that oxygen vacancies contribute to the nucleation and activation of dislocation in SrTiO₃. In this

point of view, our calculations agree well with the recently published experimental results that SrTiO₃ with higher vacancy concentration favors dislocation nucleation [Stich S, et al. J Am Ceram Soc. 2021, 105: 1318-29; Fang X, et al. Scripta Mater 2020, 188: 228-32]. The microscopic mechanism is still an open question in the published papers, which is successfully elucidated in our work.

In order to evidence the calculated result that oxygen vacancies can reduce the Peierls stress in SrTiO₃, we performed a compression test on the {100} SrTiO₃ crystals before and after reduction treatment. The stress-plastic strain curves are shown in the figure below (Fig. S6). The plastic strain can be divided into four stages of 0, I, II and III, which agree with the previous report [Yang K H et al., J. Am. Ceram. Soc. 2011, 94: 3104-3111]. The critical resolved shear stress required by the activation of <110>{110}-type dislocations can be calculated by the measured yield stress times the Schmid factor (~0.5 for <110>{110}-type dislocations). The measured yield stress is about 120 MPa for the intrinsic SrTiO₃ crystals, which agrees well with the previous report [Brunner D et al., Acta Mater. 2006, 54: 4999-5011]. The yield stress decreases to 77 MPa after the oxygen reduction treatment, indicating that oxygen reduction can lower the critical shear stress. It proves our calculated result that oxygen vacancies contribute to decreasing the Peierls stress of <110>{110}-type dislocations.

Fig. S6 Stress-plastic strain curves of the {100} SrTiO₃ crystals before and after oxygen reduction.

The investigation on the relation between oxygen vacancy and dislocation motion requires a dynamic simulation on the interaction between oxygen vacancies and a moving dislocation. The MD simulation in our previous manuscript indicates that oxygen vacancies may help with kink formation, but it's hard to record the migration process of the kink. In order to analyze the effect of oxygen vacancy on dislocation motion, we recorded the migration process of dislocation line on as shown in the figure below (Fig. 5). A single oxygen vacancy has no influence on the glide of dislocation line, but a cluster of three oxygen vacancies shows a pinning effect on the dislocation, which effectively impedes the motion of dislocation line. The difference may be ascribed to the fact that the size of lattice strain near an oxygen vacancy is too small to affect the motion of dislocation line.

Fig. 5 the migration of dislocation line on the glide plane with a single oxygen vacancy (left figure) and a cluster of three oxygen vacancies (right figure).

The discussion on the effect of dislocations on toughness has been modified following the reviewer's suggestion. According to the reference [Porz L, et al. Dislocation-toughened ceramics. Mater Horiz. 2021, 8: 1528-37], crack propagation is easier than the nucleation of dislocations in ceramics. The great mobility of $\langle 110 \rangle \{ 110 \}$ type dislocations is unfortunately useless for increasing the toughness of SrTiO_3 because it is difficult for dislocations to nucleate near crack tips before the cracks extend. According to our calculations, the oxygen vacancies in SrTiO_3 can reduce the average misfit energy and the Peierls stress of $\langle 110 \rangle \{ 110 \}$ dislocation, contributing to the nucleation and activation of dislocation. Thus, the oxygen vacancies are in favor of preventing crack extension, which is evidenced by the indentation tests. As reviewed by the reviewer, the Vickers indentation fracture toughness needs to be used cautiously. However, it still can represent some form of a crack arrest phenomenon. We added the Vickers hardness H_v and crack length c in Fig. 6c to support the analysis on the fracture toughness.

Fig. 6(c) Indentation fracture toughness K_{IC} , Vickers hardness H_v and crack length c measured in the indentation test.

Revision_2: The discussions on the dynamic process of dislocation have been modified in page 16, paragraph 3. The compression test on SrTiO_3 crystals and the relative discussions have been added in page 17, paragraph 2 of article and page 6 of the supporting materials. The discussions on dislocation motion are not related to the LME model proposed in our manuscript. Therefore, we decided to move the detailed MD simulations into the supporting materials (pages 8-10) as extended contents, and only keep the main conclusions and Fig. 5 in the manuscript (page 17, paragraph 1). The

effect of dislocations on toughness has been modified in page 18, paragraph 1 and page 19, paragraph 1 of article.

Comment_3: Please give the Peierls-Nabarro equation for reference.

Response_3: The Peierls-Nabarro equation has been added in page 4, paragraph 1 of article.

Comment_4: Figure 6 needs a scale bar

Response_4: The scale bar has been added in the figure.

Comment_5: Figure 1: The arrows drawn indicate a uniaxial stress but not a shear stress, please correct.

Response_5: The illustration “shear direction” in Figure 1 has been replaced by “uniaxial displacement”.

Comment_6: Disregistry. What does this word mean?

Response_6: Disregistry means atomic misfit. The word has been used in several publications about dislocation modelling [Joós, B. et al. Phys. Rev. B 1994, 50: 5890-5898; Joós, B. et al. Phys. Rev. Lett. 1997, 78: 266-269; Ferré, D. et al. Phys. Rev. B 2008, 77: 014106].

Revision_6: This word may be unfamiliar to readers, so we have replaced it by the phrase “atomic misfit”.

Comment_7: Figure 3b has the unit dislocation density in the surrounding of a dislocation on the y-axis. Dislocation density quantifies the number of dislocations in a volume/area. The unit does not appear to make sense here.

Response_7: In the Peierls-Nabarro model, the dislocation core is regarded as a continuous distribution of shear $S(x)$ or infinitesimal dislocations with density $\rho(x)$ [$\rho(x) = dS(x)/dx$], where x is the coordinate in the glide plane along the direction normal to dislocation line. The dislocation density $\rho(x)$ in Figure 3b refers to the density of

infinitesimal dislocations that distribute along the x-direction.

Revision_7: The explanation on $\rho(x)$ has been added in the captions of Figs. 3 and 4.

Comment_8: While language is generally okay, there are multiple incidents where language is not at 100%. E.g. line 317 says “Fig. 4c shows the distribution of dislocation core.” This seems incomplete or grammatically incorrect. A careful revision of the language (maybe with editorial assistance) may remedy this. Also, small things such as line 454 “data availability” not “date availability”.

Response_8: The manuscript has been carefully polished. The sentence “Fig. 4c shows the distribution of dislocation core.” has been modified as “Fig. 4c shows the x-dependence of $S(x)$ and $\rho(x)$.” The spelling mistake “date availability” has been revised. Other revised grammatical and spelling mistakes have been marked in the manuscript.

Comment_9: Line 101: experimental values are cited with ref. 18 (Nabarro 1947). Please provide the experimental value and try to find a reference specific to SrTiO₃ so that your derived value can be better compared.

Response_9: We are sorry for this mistake. The experimental values should be cited with ref. 3 [Brunner, D. Low-temperature plasticity and flow-stress behaviour of strontium titanate single crystals. *Acta Mater.* **54**, 4999-5011 (2006)] instead of ref. 18. We have also checked the reference numbers of other citations.

Comment_10: Line 247. How can only half a b-vector be observed? Is this a statistically representative case? E.g. Jin et al. (TEM study of < 110 >-type 35.26 degrees dislocations specially induced by polishing of SrTiO₃ single crystals) and others found that partial dislocations only dissociated about 5 unit cells.

Response_10: The HRTEM image is not clear near another partial dislocation, so we overlapped the HRTEM image with its Fourier filtered image in the area encircled by the white rectangle. As shown in the figure below, this partial dislocation is marked by the blue circle. The separation of the two parts is about 2.6 nm.

Fig. 3(d) high-resolution transmission electron microscopy (HRTEM) image containing two partial $\langle 011 \rangle\{011\}$ -type dislocations from the [100] perspective. The Burgers circle yields a Burgers vector $\frac{1}{2}[011]$ of the left partial dislocation. The HRTEM image near another partial dislocation is not clear, so the area, encircled by the white rectangle, overlaps its Fourier filtered image. The right partial dislocation is marked by the blue circle.

Revision_10: The Fig. 3d has been replaced by the figure above. The discussion on the two partial dislocations has been added in page 13, paragraph 1 of article and the caption of Figure 3d.

Reviewer #3 (Remarks to the Author)

The submitted manuscript deals with the estimation of the lattice resistance to dislocation motion in SrTiO_3 with and without the presence of an oxygen vacancy. The authors propose a new way of calculating the Peierls stress which increases the previous estimates from very few to hundreds of MPa, with the latter much closer to experimental observations.

Overall, I agree with the authors that this is an interesting and still very poorly understood topic and that new approaches that take into account the peculiarities of ordered crystals are needed.

However, I do have several reservations about the presented hypotheses, data from the literature and mechanisms.

Response_0: Thanks for the evaluation and the comments. An in-depth understanding of the dislocation motion process in non-metallic materials becomes increasingly important. Our work attempts to address this issue by proposing a local-misfit-energy (LME) method based on the Peierls-Nabarro theory. The method can achieve a good estimate of Peierls stress, and more importantly, provides detailed physical insights into the Peierls plasticity of SrTiO₃, such as the re-bonding process of the Ti-O bonds and the reversal of lattice dipoles, which are lacked in the traditional GSF method. In addition, our theory provides a practical approach to calculate the Peierls stress of the complex model accompanied with defects, which the other methods fail to handle. We have updated our view on the dynamic process of dislocation based on the reviewers' constructive suggestions. New simulations and experiments are added to support our analysis. The comparisons between our work and some traditional methods mentioned by the reviewer, such as the Peierls-Nabarro-Galerkin method and the nudged elastic band approach, have been added.

Comment_1: the main assumption, i.e. that in the literature the Peierls stress is solely estimated based on a GSF method that does not take into account any relaxation normal to the slip plane, is not a complete view of the literature. For example, Cordier, Carrez et al. have calculated the Peierls stresses of several complex crystals including perovskites and using GSFs on different crystal planes can achieve a good estimate of the core structure and Peierls stress. Among their work there is also an estimate of 350 MPa for the Peierls stress of SrTiO₃ (although they did estimate much much less before using a 1D estimate), which is very close to what the authors suggest here. An example of this work can be found here: <https://doi.org/10.1016/j.scriptamat.2010.04.045>.

Response_1: Thanks for the comment. We didn't assume that the GSF method does not take into account any relaxation normal to the slip plane in literature. On the contrary, our main assumption is that the GSF method only takes into account relaxation normal to the slip plane, but the degrees of freedom parallel to the slip plane are constrained. Accordingly, the GSF energy only includes the inelastic strain energy. However, in the PN model, the restoring force originates from the strain energies

including both elastic and inelastic parts. The ignorance of the elastic part makes the GSF approach fail to account for the bond breaking and re-bonding process in non-metallic materials, resulting in an inaccurate physical picture of the dislocation motions. Our method introduces the in-plane freedom degrees for the atoms near the shear plane, so the misfit region is no longer restricted to the shear plane and the elastic energy can be accurately taken into account.

This literature referred by the reviewer [doi: 10.1016/j.scriptamat.2010.04.045] applied a so-called Peierls–Nabarro–Galerkin (PNG) method to calculate the Peierls stress of SrTiO₃. In the PNG model, the atoms are still only allowed to relax normal to the slip plane. However, it tries to evaluate the elastic strain energy E^e by introducing a three-dimensional displacement field u , which is computed by an element-free Galerkin method. The slip plane is determined by minimizing the following energy ε :

$$\varepsilon = \int_v \left\{ E^e[u, f] - \frac{1}{2} \Omega \dot{u}^2 \right\} dV + \int_\Sigma E^{isf}[f] d\Sigma$$

where f is a two-dimensional field which is expressed in the normal basis of the slip plane, and E^{isf} is the inelastic stacking fault energy from which all the linear elastic part has been subtracted. As the authors mentioned in the literature, the f field, which provides a displacement jump when crossing the slip plane of dislocation, is purely along $\langle 110 \rangle$. In another word, the slip system of dislocation is the same as their 1D estimate [Phys. Rev. B, 2008, 77: 014106]. Therefore, the major amelioration in the PNG model is to take the elastic strain energy into account. The calculated energy ε is beyond the GSF energy, which leads to a better estimate of 350 MPa for the Peierls stress.

However, Carrez et al. estimated the displacement field u based on an element-free Galerkin method (an approximation of a continuous field representation), which is a numerical algorithm and do not explicitly consider the atomic scale details of dislocation cores. Although the equivalent of inelastic energy can be roughly estimated, the physical mechanism of the slip process that are closely related with the atomic structure of SrTiO₃ cannot be provided. Besides, the PNG method is still based on the GSF model and uses the GSF energy as an input [Denoual C, Phys. Rev. B 2004, 70:

024106]. Therefore, the PNG method has the same problem with the traditional GSF approach that it cannot treat dislocation structures with impurities or defects. In contrast, the estimation of the total strain in our manuscript is based on the first principles calculation. The simulation of atomic displacements is closely related to the properties of SrTiO₃ as we analyzed in the manuscript. Our method can achieve a good estimate of Peierls stress, and more importantly, provides unexplored physical insights into the Peierls plasticity of SrTiO₃, such as the re-bonding process of the Ti-O bonds and the reversal of lattice dipoles, which is one of the main targets of our manuscript. In addition, our theory provides a practical approach to deal with the complex model accompanied with defects, such as oxygen vacancies, which has been a pending issue for many years.

Revision_1: The comparison with the PNG method has been added in page 5, paragraph 2 of article.

Comment_2: there are also other methods which allow the estimation of energy barriers in deformation of complex crystals and which account for the fact that dislocation motion may not be planar or happen along a rigid shear direction. This is the case for example for synchroshear, which was first suggested for Al₂O₃ but is mainly associated with Laves phases. An example may be found here: <https://www.sciencedirect.com/science/article/pii/S1359646219301599>.

Response_2: According to the literatures [Guérolé J, et al. Scripta Mater. 2019, 166: 134-138; Zhang W, et al. Phys. Rev. Lett. 2011, 106: 165505], the simulation of synchroshear deformation is performed with the nudged elastic band (NEB) approach. The purposes of the NEB method and our work are totally different. The NEB method is applied to find the slip path of dislocation, but our method based on the PN theory aims to calculate the Peierls stress and the structure of dislocation core. Besides, the two methods are based on the different models, so the energy barriers calculated in the NEB approach cannot be used to solving the Peierls-Nabarro equation. Accordingly, the calculations of the Peierls stress and the structure of dislocation core are beyond the ability of the NEB approach. The NEB method may be good at finding the slip path of

dislocation, but it is not applicable to investigate the activation and nucleation of dislocation, which are important to the mechanical toughness especially for ceramic materials (the details on dislocation and toughness are shown in Response 3).

Revision_2: The comparison with the synchroshear has been added in page 5, paragraph 2 of article.

Comment _3: the authors plot in figure 5 and discuss in the text the acceleration of kink formation in the presence of a vacancy. However, there is no discussion as to what happens as the dislocation line travels beyond the vacancy - pinning is also an option, which may result in the opposite effect overall and I would suggest to discuss this explicitly, hoping that the available data allows this directly already.

- it is not quite clear why the authors relate dislocation mobility primarily to the more indirect measure of crack propagation, although slip bands are clearly indicated in their experimental figures. Is this as the surface obscures slip bands more completely after reduction? Why discuss the measured hardness directly. there is quite a bit of work on SrTiO₃, its dislocations and also the role of vacancies in these works that carry out a very careful dislocation etchpit analysis:
<https://doi.org/10.1016/j.scriptamat.2020.07.033> & <https://doi.org/10.1111/jace.18118>.

Response_3: We have corrected our opinions on fracture toughness, dislocation motion and the effect of oxygen vacancies. The effect of oxygen vacancies on dislocation should be divided into three stages: nucleation, activation and motion. The misfit energy and the Peierls stress are the criterions for nucleating and activating a dislocation. It is improper to use them to analyze dislocation motion as we did. In our calculation, the average misfit energy [$W_{ave} = \frac{1}{a'} \int_0^{a'} W(u) du$] and the Peierls stress of the dislocation decrease after introducing oxygen vacancies in SrTiO₃. The results indicate that oxygen vacancies contribute to the nucleation and activation of dislocation in SrTiO₃. In this point of view, our calculations agree well with the recently published works on SrTiO₃, its dislocations and the role of vacancies [Stich S, et al. J Am Ceram Soc. 2021, 105: 1318-29; Fang X, et al. Scripta Mater 2020, 188: 228-32], and provide

a theoretical explanation for their conclusion that SrTiO₃ with higher vacancy concentration favors dislocation nucleation.

In order to evidence the calculated result that oxygen vacancies can reduce the Peierls stress in SrTiO₃, we performed a compression test on the {100} SrTiO₃ crystals before and after reduction treatment. The stress-plastic strain curves are shown in the figure below (Fig. S6). The plastic strain can be divided into four stages of 0, I, II and III, which agree with the previous report [Yang K H et al., J. Am. Ceram. Soc. 2011, 94: 3104-3111]. The yield stress corresponds to the beginning of plastic deformation. The critical resolved shear stress required by the activation of <110>{110}-type dislocations can be calculated by the measured yield stress times the Schmid factor (~0.5 for <110>{110}-type dislocations). The measured yield stress is about 120 MPa for the intrinsic SrTiO₃ crystals, which agrees well with the previous report [Brunner D et al., Acta Mater. 2006, 54: 4999-5011]. The yield stress decreases to 77 MPa after the oxygen reduction treatment, indicating that oxygen reduction can lower the critical shear stress. It proves our calculated result that oxygen vacancies contribute to decreasing the Peierls stress of <110>{110}-type dislocations.

Fig. S6 Stress-plastic strain curves of the {100} SrTiO₃ crystals before and after oxygen reduction.

The investigation on the relation between oxygen vacancy and dislocation motion requires a dynamic simulation on the interaction between oxygen vacancies and a

moving dislocation. The MD simulation in our previous manuscript indicates that oxygen vacancies may help with kink formation, but it's hard to record the migration process of the kink. In order to analyze the effect of oxygen vacancy on dislocation motion, we recorded the migration process of dislocation line on as shown in the figure below (Fig. 5). A single oxygen vacancy has no influence on the glide of dislocation line, but a cluster of three oxygen vacancies shows a pinning effect on the dislocation, which effectively impedes the motion of dislocation line. The difference may be ascribed to the fact that the size of lattice strain near an oxygen vacancy is too small to affect the motion of dislocation line. The discussions on dislocation motion are not related to the LME model proposed in our manuscript. Accordingly, we decided to move the MD simulations into the supporting materials as extended contents, and only keep the main conclusions and the figure below in the manuscript.

Fig. 5 the migration of dislocation line on the glide plane with a single oxygen vacancy (left figure) and a cluster of three oxygen vacancies (right figure).

The discussion on the effect of dislocations on toughness has been modified following the suggestions of Reviewer2#. According to the reference [Porz L, et al. Dislocation-toughened ceramics. *Mater Horiz.* 2021, 8: 1528-37], crack propagation is easier than the nucleation of dislocations in ceramics. The great mobility of $\langle 110 \rangle \{ 110 \}$ type dislocations is unfortunately useless for increasing the toughness of SrTiO_3 because it is difficult for dislocations to nucleate near crack tips before the cracks extend. According to our calculations, the oxygen vacancies in SrTiO_3 can reduce the average misfit energy and the Peierls stress of $\langle 110 \rangle \{ 110 \}$ dislocation, contributing to the nucleation and activation of dislocation. Thus, the oxygen vacancies are in favor of preventing crack extension, which is evidenced by the indentation tests. We added the Vickers hardness H_v and crack length c in the figure as shown below (Fig. 6c) to support the analysis on the fracture toughness.

Fig. 6(c) Indentation fracture toughness K_{IC} , Vickers hardness H_v and crack length c measured in the indentation test.

The Vickers hardness of SrTiO_3 crystals almost remain the same after oxygen reduction as show in Fig. 6c, which can be ascribed to two competing effects of oxygen vacancies. A lower mobility of dislocation can reduce the size of plastic zone produced by the indentation load, leading to a larger magnitude of hardness [Matyunin VM et al., *IOP Conf. Ser.: Mater. Sci. Eng.* 2019, 537: 032004]. As mentioned above, oxygen vacancies can reduce the dislocation mobility of SrTiO_3 , and accordingly have a strengthening effect on hardness. Moreover, hardness generally shows a positive

correlation to elastic modulus [Bao WY et al., *Acta Mater.* 2004, 52: 5397-5404]. The elastic moduli of ABO_3 perovskites usually decrease after oxygen reduction [Hoedl M F et al., *Acta Mater.* 2018, 160: 247-256]. Thus, oxygen vacancies also have a weakening effect on the hardness of $SrTiO_3$. The two competing effects of both strengthening and weakening make the hardness of $SrTiO_3$ has little change after oxygen reduction treatment. The competing effects of oxygen vacancies on hardness have also been reported in cerium oxide [Wang YL et al., *ECS Trans.* 2006, 1: 23-31].

Revision_3: The discussions on the dynamic process of dislocation have been modified in page 16, paragraph 3. The compression test on $SrTiO_3$ crystals and the relative discussions have been added in page 17, paragraph 2 of article and page 6 of the supporting materials. The discussions on dislocation motion are not related to the LME model proposed in our manuscript. Therefore, we decided to move the MD simulations into the supporting materials (pages 8-10) as extended contents, and only keep the main conclusions and the Fig. 5 in the manuscript (page 17, paragraph 1). The effect of dislocations on toughness has been modified in page 18, paragraph 1 of article. The discussion on Vickers hardness has been added in page 19, paragraph 1 of article.

Reviewers' comments:

Reviewer #1 (Remarks to the Author):

The authors have done substantial work for the revision of the manuscript, and in many places, the changes are in my view, appropriate. Overall, the work represents a good advancement in the field and is expected to have a high impact. Therefore, I recommend it be published in Nature Communications.

Reviewer #3 (Remarks to the Author):

Thank you for the very detailed responses and the additional work and wording which was done to strengthen the conclusions of the manuscript.

I would be happy for this manuscript to be published.

Reviewer #4 (Remarks to the Author):

The fundamental understanding of dislocation dynamics in oxides is of urgent interest and importance, particularly in light of the recent upsurge in the dislocation mechanics and functionality studies in such materials that exhibit dislocation plasticity (even at room temperature), which make such materials hold potential for technological applications. This work proposes a new simulation protocol using "local misfit energy" to be distinguished from GSF energy to address the Peierls energy for dislocation behavior in SrTiO₃ (STO). Although the attempt merits its value, however, due to the following critical aspects, the reviewer cannot recommend this work for publication, not in the current journal nor in other scientific journals in its present form:

1. The room-temperature dislocations in single-crystal STO are dominantly screw type, not edge type, as has been extensively reported in the literature, see e.g., Brunner et al., *Acta Mater.*, 2006 (the authors were also citing this paper for comparing the CRSS with their simulation in Fig. S4 in the Rebuttal letter, which makes little sense to the reviewer). The current simulation focuses on the edge type and attempts to extend the conclusion to the dislocation dynamics at room temperature in STO. This could be a fundamentally misleading effort and the authors should consider extending their approach to screw-type dislocation.

It is worth mentioning that, edge dislocations can be dominating at high temperatures, and the simulation effort in the current work still has its merit and the authors may consider a such extension to high-temperature slip system in STO in their future study.

2. Following point 1, screw-type dislocations in STO tend to cross slip and greatly promote the dislocation multiplication hence dislocation density, and this process is way more important than dislocation nucleation and motion in this case, in order to promote the plasticity in STO at room temperature. Similar results are also readily available in the textbook knowledge on LiF (same cubic structure as STO) regarding dislocation plasticity. See the book by Hull & Bacon, *Introduction to Dislocations*.

3. It is appreciated that the authors attempted to validate their simulation prediction in experimental tests, particularly related to the Vickers hardness and fracture toughness in reference and reduced STO samples. However, such an attempt is poorly justified based on the following points:

- 3.1. Besides dislocation nucleation and motion, the authors ignored (or maybe were not aware of) the dislocation multiplication in point 2. During the Vickers indentation process (as the Indenter is pressing into the material), consider the dislocation generation, it will be dislocation nucleation, multiplication, while being accompanied by movement of the mobile dislocations. These processes can be greatly modified/affected by the defect chemistry (e.g., oxygen vacancy concentration) of the samples.

- 3.2. Consider the scenario that higher oxygen vacancy concentration in the reduced samples could have enhanced the dislocation multiplication (which is likely possible), the total input energy from the indentation will be largely dissipated by the generation of more dislocations in the reduced samples,

hence reducing the energy available for crack formation (hence crack length could decrease). This point must be considered and checked before one could assess the crack length and indentation fracture toughness using such a method. In combination with the 2nd reviewer's 2nd comment (regarding crack tip dislocation nucleation, etc.) in the last round review, the authors should have a more complete picture now for the deformation process.

3.3. The authors mentioned the term "toughness" and "fracture toughness" several times in the manuscript. These two terms are completely different and must be clarified. Fracture toughness refers to the resistance to crack propagation, while toughness can be described by the area under the stress-strain curves. In this respect, the toughness of the reduced sample is much smaller than the reference sample before reduction treatment (Fig. S6), while the authors are trying to prove the reduced sample was supposed to display higher fracture toughness (K_{Ic} in this case). Please check these basic concepts in mechanics textbooks.

4. The authors discussed the oxygen vacancy on the dislocation nucleation, which also agrees with the most recent experimental results such as by Stich et al. J. Am. Ceram. Soc 2022 and Fang et al, Scripta Mater., 2020 as mentioned by the authors. These are very encouraging matches between simulation and experiments. However, it should be pointed out that these works used nanoindentation tests to probe the dislocation nucleation in a very small stressed volume. In the current work, the bulk uniaxial compression tests were performed to validate the oxygen vacancies effect on the Peierls stress: the authors should be aware that the yielding (incipient plasticity in bulk sample) in this case is mostly achieved by dislocation multiplication and motion process (see point 3 above), while dislocation nucleation is not relevant anymore.

It is also very alarming to the reviewer that the authors cut the samples first into 2x2x4 mm³ in size and then performed the compression tests: cutting STO samples would immediately induce many surface dislocations (penetrating into the surface by about several micrometers) and these surface dislocations are effective sources to multiply many more new dislocations to promote the plasticity. Plus that the reduction at 1450C for 6 hours would also anneal some surface dislocations, making the two samples have different initial states (reference sample with surface dislocations from cutting and without thermal treatment, and reduced sample with thermal treatment and most likely a different surface dislocation structure). The careful treatment of these cut samples (e.g., to polish away the surface cut induced regions) to remove such surface dislocations must be added to make sure the samples do have similar initial dislocation conditions. The surface dislocations by cutting can be easily checked by the optical microscope to see abundant slip traces.

5. Regarding the reducing treatment in Argon gas, although the authors measured the electrical resistivity to suggest a higher oxygen vacancy, a most robust analysis should be performed using impedance spectroscopy, which is rather a standard technique for such purposes.

6. The overall experimental section seems very weak and poorly defined to the reviewer, and the authors are suggested to check the literature with experimental details in recent years or consult experimental experts to avoid such experimental pitfalls.

Response Letter

We appreciate the constructive and helpful suggestions from the new reviewer and we have revised the manuscript accordingly. The changes are marked in the revised manuscript, and the responses to the comments are listed as follows:

Reviewer #4 (Remarks to the Author)

Comment_0: The fundamental understanding of dislocation dynamics in oxides is of urgent interest and importance, particularly in light of the recent upsurge in the dislocation mechanics and functionality studies in such materials that exhibit dislocation plasticity (even at room temperature), which make such materials hold potential for technological applications. This work proposes a new simulation protocol using “local misfit energy” to be distinguished from GSF energy to address the Peierls energy for dislocation behavior in SrTiO₃ (STO). Although the attempt merits its value, however, due to the following critical aspects, the reviewer cannot recommend this work for publication, not in the current journal nor in other scientific journals in its present form.

Response_0: Thanks for your evaluation and the valuable comments. We have responded to the reviewer’s concern on the screw-type dislocation in Response_1 and 2. With the help of the reviewer’s suggestions, the discussions in the experimental section have been strengthened and new experiments have been added. All the comments have been addressed point by point.

Comment_1: The room-temperature dislocations in single-crystal STO are dominantly screw type, not edge type, as has been extensively reported in the literature, see e.g., Brunner et al., *Acta Mater.*, 2006 (the authors were also citing this paper for comparing the CRSS with their simulation in Fig. S4 in the Rebuttal letter, which makes little sense to the reviewer). The current simulation focuses on the edge type and attempts to extend the conclusion to the dislocation dynamics at room temperature in STO. This could be

a fundamentally misleading effort and the authors should consider extending their approach to screw-type dislocation. It is worth mentioning that, edge dislocations can be dominating at high temperatures, and the simulation effort in the current work still has its merit and the authors may consider a such extension to high-temperature slip system in STO in their future study.

Response_1: Thanks for your comment. We notice that the major concern of the reviewer is that the room-temperature dislocations in single-crystal STO are dominantly screw-type, and our simulation on edge dislocations does not accord with the experimental fact. However, we did a thorough literature survey on the dislocation type of STO at room temperature. We found all the published papers clearly conclude that the room-temperature plasticity of single-crystal STO is dominantly governed by edge dislocations, not screw dislocations. For example, in the literature mentioned by the reviewer, Brunner et al. [Acta Mater. 2006, 54: 4999-5011. Page 5011, paragraph 2: “The ductility of strontium titanate was measured below RT down to 42 K... the high-temperature regime dislocations of rather edge type govern the plastic behavior while in the low-temperature regime dislocations of rather screw type”] have concluded that the edge dislocations govern the plastic behavior of STO from room temperature down to 225 K, and the screw dislocations play the dominating role at low temperatures. They also pointed out in another literature [W. Sigle et al., Philos. Mag., 2006, 86: 4809-4821. Page 4812, paragraph 3] that “after 115 K deformation dislocations are predominantly of screw type whereas this is much less pronounced for room temperature deformation”.

More recently, Yang et al. [J. Am. Ceram. Soc. 2011, 94: 3104-3111] have reported the microscopic dislocation substructure of STO single-crystal at different plastic deformation stages. They divided the plastic deformation process into four stages as shown in Fig. R1. According to the literature [J. Am. Ceram. Soc. 2011, 94: 3104-3111. Page 3107, paragraph 1: “The representative deformation microstructure at point (i) of stage 0 consists of interacting dislocations, as shown in Fig. 4(a)... Dislocations shown in Fig. 4 are predominantly of the pure edge type”], the dislocations are predominantly of the pure edge type in the first stage (stage 0). The beginning of

this stage is directly related to the Peierls stress, which is the main target in our manuscript. Actually, few screw dislocations are observed even in the early stage II [J. Am. Ceram. Soc. 2011, 94: 3104-3111. Page 3107, paragraph 5: “Such dislocation bundles in monotonically deformed metals were often addressed as braids; here they are consisted of higher density dislocations of the pure edge type with $b= [01\bar{1}]$ and $[011]$ lying in two perpendicular planes, as indicated in Fig. 7. However, in comparison, one of the two walls (as indicated) has significantly lower dislocation density. Unlike in stage I, these dislocations form dipoles instead of dissociating into partials, as indicated. Cells contain both mixed and screw dislocations.”].

Fig. R1 Stress-strain curve (thick curve) of the $\{100\}$ SrTiO_3 crystals reported by Yang et al. [J. Am. Ceram. Soc. 2011, 94: 3104-3111].

Besides, L. Porz et al. also analyzed the dislocation structure of slip bands in uniaxially compressed SrTiO_3 single crystals [Mater. Horiz., 2021, 8: 1528]. In the literature, page 1530, paragraph 2, “In regions close to the slip band front with low dislocation density, edge-type dislocations on $\{1\bar{1}0\}$ planes with $b= [110]$ Burgers vector are predominantly found... Further behind the tip, the dislocation structure becomes much more complex and reveals more screw and mixed components (Fig. 1d).” The regions close to the tip of slip band correspond to the early stage of plastic deformation, and those further behind the tip correspond to the later stage of plastic deformation. The results also verify Yang et al.’s conclusions.

In summary, edge dislocations dominate the plasticity of STO at room temperature

and screw dislocations are hardly observed until the later stage of plastic deformation. The major characteristics of dislocations in STO, including the dislocation structure, average misfit energy and Peierls stress, which are also the main target of our paper, are all determined by the earlier stage of plastic deformation and the edge dislocations. Therefore, we believe our study on the edge dislocations in STO accords with the experimental fact and can support the main conclusions of this manuscript.

Of course, we should have compared our calculated Peierls stress with the experimental data of edge dislocations, instead of those at the low temperatures (below 150 K) that are mainly governed by screw dislocations. Thanks for pointing out the inappropriateness in Fig. S4.

Revision_1: The extrapolation of CRSS is recalculated by fitting the experimental data near room temperature. The plastic behavior is mainly governed by edge-type dislocations in this temperature range. The fitting curve in Fig. S4 has been renewed as shown below. The extrapolated CRSS is 290 MPa at 0 Kelvin, which agrees well with the calculated Peierls stress value (305 MPa) in our manuscript.

Fig. S4 Extrapolation of CRSS calculated by fitting the experimental data corresponding to $\langle 110 \rangle \{110\}$ edge dislocation [D. Brunner, Acta Mater. 2006, 54: 4999-5011].

Comment_2: Following point 1, screw-type dislocations in STO tend to cross slip and greatly promote the dislocation multiplication hence dislocation density, and this

process is way more important than dislocation nucleation and motion in this case, in order to promote the plasticity in STO at room temperature. Similar results are also readily available in the textbook knowledge on LiF (same cubic structure as STO) regarding dislocation plasticity. See the book by Hull & Bacon, Introduction to Dislocations.

Response_2: As we demonstrated in Response_1, the plasticity of STO at room temperature is governed by edge-type dislocations. The screw-type dislocations play the major part at low temperatures. The cross slip and multiplication of screw-type dislocations are interesting and valuable topics, but beyond the scope of this manuscript. Although LiF also has the same cubic structure as STO, the cross-glide of screw dislocations in LiF has a dominating effect at room temperature [W. G. Johnston et al., J. Appl. Phys., 1960, 31: 632], which is different from STO.

Comment_3: It is appreciated that the authors attempted to validate their simulation prediction in experimental tests, particularly related to the Vickers hardness and fracture toughness in reference and reduced STO samples. However, such an attempt is poorly justified based on the following points:

3.1. Besides dislocation nucleation and motion, the authors ignored (or maybe were not aware of) the dislocation multiplication in point 2. During the Vickers indentation process (as the Indenter is pressing into the material), consider the dislocation generation, it will be dislocation nucleation, multiplication, while being accompanied by movement of the mobile dislocations. These processes can be greatly modified/affected by the defect chemistry (e.g., oxygen vacancy concentration) of the samples.

3.2. Consider the scenario that higher oxygen vacancy concentration in the reduced samples could have enhanced the dislocation multiplication (which is likely possible), the total input energy from the indentation will be largely dissipated by the generation of more dislocations in the reduced samples, hence reducing the energy available for crack formation (hence crack length could decrease). This point must be considered and checked before one could assess the crack length and indentation fracture toughness

using such a method. In combination with the 2nd reviewer's 2nd comment (regarding crack tip dislocation nucleation, etc.) in the last round review, the authors should have a more complete picture now for the deformation process.

Response_3.1 and 3.2: Thanks for reminding us about the significance of dislocation multiplication. It is necessary to consider the dislocation multiplication in our Vickers indentation tests. According to the literature reported by Fang et al. [Crystals 2020, 10: 933], the dominating dislocation activities related to plasticity deformation are different at the nano-, micro- and macro-scales. New dislocations are generated mainly via the nucleation process at the nano-scale, and the multiplication process occurs at the macro-scale. The dislocation nucleation and multiplication are both important at the micro-scale. Thus, the plasticity behavior is governed by a combination of dislocation nucleation, motion and multiplication in the Vickers indentation tests. As discussed in the manuscript, the oxygen vacancies contribute to the dislocation nucleation. When the newly-nucleated dislocations are activated to move, they can serve as the source of the dislocation multiplication and further improve the dislocation content. Besides, the vacancies with a large concentration also can facilitate the dislocation multiplication through the Frank-Read mechanism [F. Frank, Discuss. Farad. Soc. 1957, 23: 122-127; Z.-W. Chen, Nat. Commun. 2017, 8: 13828]. However, it should be noted that the dislocation multiplication is a dynamic process during the movement of dislocations, and its mechanism is usually complicated. The effect of oxygen vacancies on the dislocation multiplication in STO is still an open issue, which requires further efforts in both experiments and molecular dynamics simulations.

Revision_3.1 and 3.2: The dislocation multiplication has been added to complete the picture of plastic deformation process. The relevant discussions have been added in page 16, paragraph 3: "the effects of oxygen vacancies on the dislocations in SrTiO₃ can be divided into several stages: nucleation, activation, motion and multiplication.", page 18, paragraph 1: "A large number of dislocations is required to effectively achieve the shielding effect. However, crack propagation is easier than the nucleation and multiplication of dislocations in ceramics⁵². The dislocations are difficult to nucleate or multiply near crack tips before the cracks extend... When the newly-nucleated

dislocations are activated to move, they can serve as the source of the dislocation multiplication and further improve the dislocation content.”, page 18, paragraph 2: “The micro-indentations are performed on the reduced and un-reduced SrTiO₃ crystals to prove the effect of oxygen vacancies. According to the literature reported by Fang et al.⁵³, the dominating dislocation activities include the dislocation nucleation, motion and multiplication in the micro-scale indentation tests...” and page 19, paragraph 1: “It should be noted that the dislocation multiplication is a dynamic process during the movement of dislocations, and its mechanism is usually complicated. The effect of oxygen vacancies on the dislocation multiplication in SrTiO₃ is still an open issue.”

3.3. The authors mentioned the term “toughness” and “fracture toughness” several times in the manuscript. These two terms are completely different and must be clarified. Fracture toughness refers to the resistance to crack propagation, while toughness can be described by the area under the stress-strain curves. In this respect, the toughness of the reduced sample is much smaller than the reference sample before reduction treatment (Fig. S6), while the authors are trying to prove the reduced sample was supposed to display higher fracture toughness (K_{Ic} in this case). Please check these basic concepts in mechanics textbooks.

Response_3.3: The term “toughness” refers to “fracture toughness” in the manuscript. Thanks for clarifying the difference between these two terms.

Revision_3.3: The term “toughness” has been modified as “fracture toughness” in the manuscript.

Comment_4.1: The authors discussed the oxygen vacancy on the dislocation nucleation, which also agrees with the most recent experimental results such as by Stich et al. J. Am. Ceram. Soc 2022 and Fang et al, Scripta Mater., 2020 as mentioned by the authors. These are very encouraging matches between simulation and experiments. However, it should be pointed out that these works used nanoindentation tests to probe the dislocation nucleation in a very small stressed volume. In the current work, the bulk uniaxial compression tests were performed to validate the oxygen vacancies effect on

the Peierls stress: the authors should be aware that the yielding (incipient plasticity in bulk sample) in this case is mostly achieved by dislocation multiplication and motion process (see point 3 above), while dislocation nucleation is not relevant anymore.

Response_4.1: We agree with the reviewer about their opinions on the nanoindentation tests and the bulk uniaxial compression tests. The former is dominated by dislocation nucleation and motion, and the latter is dominated by dislocation motion and multiplication. Therefore, the literatures reported by Stich et al. and Fang et al. are applied in the manuscript to evidence the effect of oxygen vacancies on dislocation nucleation. The bulk uniaxial compression tests are performed only for validating the oxygen vacancies' effect on the Peierls stress, which is the criterion for activating a dislocation to move. The calculated results on the dislocation nucleation and the Peierls stress were discussed in the same paragraph, which may be the reason for the reviewer's concern.

Revision_4.1: The discussions on the dislocation nucleation and the Peierls stress have been divided into page 16, paragraph 3: "In our calculation, the average misfit energy decreases after introducing oxygen vacancies, indicating that oxygen vacancies contribute to the nucleation of dislocation in SrTiO₃. The results agree well with the recently published experimental works on SrTiO₃^{47, 48}, in which the SrTiO₃ with higher vacancy concentration favors the dislocation nucleation." and page 17, paragraph 1: "The calculated Peierls stress decreases after introducing oxygen vacancies in SrTiO₃, which suggests oxygen vacancies will help to activate dislocations to move. In order to evidence this effect, the compression test on the {100} SrTiO₃ crystals before and after reduction treatment is carried out at room temperature...", respectively.

Comment_4.2: It is also very alarming to the reviewer that the authors cut the samples first into 2x2x4 mm³ in size and then performed the compression tests: cutting STO samples would immediately induce many surface dislocations (penetrating into the surface by about several micrometers) and these surface dislocations are effective sources to multiply many more new dislocations to promote the plasticity. Plus that the reduction at 1450C for 6 hours would also anneal some surface dislocations, making

the two samples have different initial states (reference sample with surface dislocations from cutting and without thermal treatment, and reduced sample with thermal treatment and most likely a different surface dislocation structure). The careful treatment of these cut samples (e.g., to polish away the surface cut induced regions) to remove such surface dislocations must be added to make sure the samples do have similar initial dislocation conditions. The surface dislocations by cutting can be easily checked by the optical microscope to see abundant slip traces.

Response_4.2: The description of sample preparation for the compression tests was too simple. Actually, we have already considered the initial states of the reduced samples and the reference samples. The samples for the compression tests were prepared by the following processes. The quadrangular prisms with dimensions $2 \times 2 \times 4 \text{ mm}^3$ were cut from the initial materials. Then, the prisms were polished in order to observe the slip lines during the compression tests, which can be regarded as a simple criterion for the stage of plastic deformation [D. Brunner, *Acta Mater.* 2006, 54: 4999-5011; K.-H. Yang et al., *J. Am. Ceram. Soc.* 2011, 94: 3104-3111]. No slip traces were observed in the polished samples, while the surface stress is also introduced by the polish process. In order to release the surface stress and gain similar initial surface conditions, the polished samples were divided into two groups. One is reduced at $1450 \text{ }^\circ\text{C}$ for 6 hours in the Argon atmosphere as the oxygen-deficient samples, and the other is also heated at $1450 \text{ }^\circ\text{C}$ for 6 hours in the air atmosphere as the reference samples.

Revision_4.2: The details on the sample preparation for the compression tests have been added in the caption of Fig. S6. The relevant instruction has been added in page 23, paragraph 3 of article: “The uniaxial compression tests were performed by using the SrTiO_3 crystals in the shape of quadrangular prisms with dimensions $2 \times 2 \times 4 \text{ mm}^3$. The details on the specimen preparation are available in the supporting materials.”

Comment_5: Regarding the reducing treatment in Argon gas, although the authors measured the electrical resistivity to suggest a higher oxygen vacancy, a most robust analysis should be performed using impedance spectroscopy, which is rather a standard technique for such purposes.

Response_5: We have measured the impedance spectroscopy (Fig. S7) of the reference sample and the reduced sample to compare their electrical conductivity, which can be regarded as a criterion for the oxygen vacancy concentration in SrTiO₃ [S. Stich et al., J. Am. Ceram. Soc. 2022, 105: 1318-1329]. Oxygen vacancies and electrons are introduced simultaneously ($O_O^\times \rightarrow V_O^{\bullet\bullet} + 2e' + 1/2O_2$) during the reduction treatment, and they can contribute to improving the electrical conductivity. The dominant electrical conduction behavior can be either electronic or ionic type, which depends on the temperature and oxygen vacancy concentration [R. A. De Souza, Adv. Funct. Mater. 2015, 25: 6326-6342]. The reduced sample has a weak frequency response as shown in Fig. S7(b), indicating that the electronic conductivity plays the dominant role. Besides, the electrical resistance of the reduced sample is too low (only a few Ohm), so we short-circuited the sample and measured the impedance again (red squares in the figure) in order to eliminate the influence of wires and electrodes. The electrical conductivity of SrTiO₃ increases from 8.22E-8 S/cm to 0.77 S/cm after the reduction treatment, indicating a high oxygen vacancy concentration in the reduced sample.

Fig. S7 Nyquist plot of impedance for (a) the reference sample (without reduction treatment) and (b) the reduced sample at 300 °C. The red squares refer to the measured data for the short-circuited sample. The insert figure of Fig. S7(b) shows the zoom-in on the scale of Z' .

Revision_5: The impedance spectroscopy of the reference sample and the reduced sample has been measured. The relevant contents have been added in page 22,

paragraph 2 of article: “Oxygen vacancies and electrons are introduced simultaneously ($O_O^\times \rightarrow V_O^\bullet + 2e' + 1/2O_2$) during the reduction treatment, and they can contribute to the improving the electrical conductivity. The impedance spectroscopy measurements are applied to compare the electrical conductivity of the reference sample and the reduced sample. The electrical conductivity of SrTiO₃ crystal increases from 8.22E-8 S/cm to 0.77 S/cm after the reduction treatment, indicating a high oxygen vacancy concentration in the reduced sample.” and in page 7 of the supporting materials to replace the previous tests.

Comment_6: The overall experimental section seems very weak and poorly defined to the reviewer, and the authors are suggested to check the literature with experimental details in recent years or consult experimental experts to avoid such experimental pitfalls.

Response_6: Thanks for the valuable comments on the experimental section. With the help of the reviewer’s suggestions, we have strengthened the discussions on the experiments. The dislocation multiplication has been added to complete the picture of the plastic deformation process. The difference between the nanoindentation tests and the bulk compression tests has been clarified. The details on the sample preparation have been added to show the initial states of the samples for the compression tests. The impedance spectroscopy has been added to analyze the concentration of oxygen vacancy.

Reviewers' comments:

Reviewer #4 (Remarks to the Author):

The Reviewer very much appreciates the Authors' efforts in systematically investigating and discussing the dislocation types, and (dominating) dislocation mechanisms, particularly at different deformation stages, as well as their new experiments in the response letter and the revised manuscript. The revisions have now clearly stated the novelty as well as its limitations (e.g., focusing on the initial stage of deformation, and not being able to address multiplication, etc.) of this work, which should be very helpful to the potential communities that will later march into this topic `dislocations in ceramics` (which is an old but also new topic). It has been a very helpful interaction between the reviewer and the authors. I am happy to recommend the publication of this work.

Comment: The Reviewer very much appreciates the Authors' efforts in systematically investigating and discussing the dislocation types, and (dominating) dislocation mechanisms, particularly at different deformation stages, as well as their new experiments in the response letter and the revised manuscript. The revisions have now clearly stated the novelty as well as its limitations (e.g., focusing on the initial stage of deformation, and not being able to address multiplication, etc.) of this work, which should be very helpful to the potential communities that will later march into this topic `dislocations in ceramics` (which is an old but also new topic). It has been a very helpful interaction between the reviewer and the authors. I am happy to recommend the publication of this work.

Response & revision: Thanks for the reviewer's comment and positive evaluation. We have added the corresponding discussion and literature citation in the main text (paragraph 1, page 6, marked in red color) to justify our model which neglects the screw-type dislocation.